# Automated evaluation of imaginary time strong coupling diagrams by sum-of-exponentials hybridization fitting

Zhen Huang,[1,2] Denis Golež,[3,4] Hugo U. R. Strand,[5] and Jason Kaye[2,6]

[1]*Department of Mathematics, University of California, Berkeley, CA 94720, USA*
[2]*Center for Computational Quantum Physics, Flatiron Institute, 162 5th Avenue, New York, NY 10010, USA*
[3]*Jožef Stefan Institute, Jamova 39, SI-1000, Ljubljana, Slovenia*
[4]*Faculty of Mathematics and Physics, University of Ljubljana, Jadranska 19, 1000 Ljubljana, Slovenia*
[5]*School of Science and Technology, Örebro University, SE-701 82 Örebro, Sweden*
[6]*Center for Computational Mathematics, Flatiron Institute, 162 5th Avenue, New York, NY 10010, USA*

We present an efficient separation of variables algorithm for the evaluation of imaginary time Feynman diagrams appearing in the bold pseudo-particle strong coupling expansion of the Anderson impurity model. The algorithm uses a fitting method based on AAA rational approximation and numerical optimization to obtain a sum-of-exponentials expansion of the hybridization function, which is then used to decompose the diagrams. A diagrammatic formulation of the algorithm leads to an automated procedure for diagrams of arbitrary order and topology. We also present methods of stabilizing the self-consistent solution of the pseudo-particle Dyson equation. The result is a low-cost and high-order accurate impurity solver for quantum embedding methods using general multi-orbital hybridization functions at low temperatures, appropriate for low-to-intermediate expansion orders. In addition to other benchmark examples, we use our solver to perform a dynamical mean-field theory study of a minimal model of the strongly correlated compound $Ca_2RuO_4$, describing the anti-ferromagnetic transition and the in- and out-of-plane anisotropy induced by spin-orbit coupling.

## I. INTRODUCTION

Strongly correlated electron systems exhibit complex quantum phenomena such as magnetism, non-Fermi liquid behavior, and metal-insulator transitions [1, 2]. Strong electron correlations defy mean-field approximations, requiring a more complete many-body treatment. Dynamical mean-field theory (DMFT) [3, 4], a quantum embedding method which maps the many-body problem to the solution of an impurity model embedded in a self-consistently determined bath, has emerged as a leading approach. The central challenge in DMFT is the full quantum many-body solution of the impurity problem.

The standard workhorse impurity solvers for DMFT are continuous time quantum Monte Carlo methods (CT-QMC) [5–7], such as the continuous time hybridization expansion (CT-HYB) [8–11] and interaction expansion (CT-INT/AUX) [6, 12] methods, which perform Monte Carlo sampling of bare diagrammatic expansions. These methods use importance sampling over all diagrammatic expansion orders, and are therefore sometimes referred to as "numerically exact." However, they have several drawbacks which limit their range of applicability, and make them too computationally expensive for use in many applications downstream of quantum embedding calculations, such as structural relaxation [13] or high-throughput materials screening. These include (1) the sign problem [7] makes them impractical for many materials-realistic systems, particularly multi-band systems with off-diagonal hybridization, e.g., spin-orbit coupling, (2) they converge at the half-order Monte Carlo rate or the first-order quasi-Monte Carlo rate [14, 15] and therefore deliver low numerical accuracy, and (3) their cost scales as $\mathcal{O}(T^{-3})$ with the temperature $T$ [16], preventing access to the low temperature regime, where

many strongly correlated collective phenomena such as superconductivity and magnetism are observed.

Moving from model systems to realistic materials simulations therefore requires more flexible and less expensive impurity solvers providing access to larger multi-band systems, off-diagonal hybridization, and lower temperatures. Fast "approximate" impurity solvers attempt to deliver sufficient accuracy with respect to the underlying impurity model without paying the price of Monte Carlo sampling over all diagrammatic orders. Several methods of this type have been proposed, including IPT [3, 17], FLEX [18, 19], SPTF [20], slave boson [21–23], slave-spin/rotor [24–27], Gutzwiller wave functions [28–32], and their extensions [33].

A straightforward approach is to truncate a diagrammatic expansion at a given order and evaluate the resulting diagrams directly. A particularly simple method of this type is the non-crossing approximation (NCA) [34–36], a first-order bold hybridization expansion, for which the self-energy requires no time integration. Its low cost has led to widespread use in complex setups such as cluster [37] or nonequilibrium extensions [38] of DMFT. For the one-crossing approximation (OCA), the second-order generalization of this approach, the self-energy involves two-dimensional time integrals, and it is therefore used less frequently [39]. Even fewer applications at third-order and beyond appear in the literature [40–42]. We refer to arbitrary-order expansions of this type as bold pseudo-particle strong coupling, or hybridization, expansions.

Including diagrams of even moderately higher order than NCA provides both additional accuracy, and a mechanism to monitor the error at a given truncation order (using an order-by-order comparison) [40, 42]. However, efficient diagram evaluation schemes are required

arXiv:2503.19727v2 [cond-mat.str-el] 8 Apr 2025

to handle the orbital sums and time integrals of increasing dimension. To this end, in Ref. 43 the authors introduced a fast imaginary time strong coupling diagram evaluation algorithm based on separation of variables. The hybridization function is approximated by a sum-of-exponentials (SOE) using the discrete Lehmann representation (DLR) [44], so that time variables can be separated and diagrammatic integrals decomposed into sequences of nested products and imaginary time convolutions, which can themselves be computed efficiently using the compact DLR basis.

We present an improved version of this algorithm, based on the following modifications: (1) We replace the DLR expansion of the hybridization function by a more tailored and efficient SOE expansion based on the AAA algorithm for rational approximation [45] and a bilevel optimization scheme [46, 47], (2) We reorganize the diagram evaluation procedure to precompute certain quantities and avoid redundant calculations, (3) We systematize the procedure in a diagrammatic language, facilitating an automated implementation for diagrams of arbitrary order and topology, and (4) We develop methods of stabilizing the self-consistent solution of the bold expansion at low temperatures. Our algorithm is several orders of magnitude faster than its predecessor [43] for the most challenging test cases considered here, leading to a practical and robust approximate impurity solver. We demonstrate its performance on benchmark examples which in previous studies were established as challenging problems for continuous time quantum Monte Carlo methods [48], namely a fermionic dimer with off-diagonal hybridization and a two-band model of correlated $e_g$ orbitals. We finally use it as an impurity solver for fully self-consistent DMFT calculations, studying the anti-ferromagnetic transition in a three-band model of $Ca_2RuO_4$ and the effect of spin-orbit coupling.

We mention a few recent methods which share similarities with our approach. The sign problem of the hybridization expansion (CT-HYB) has been addressed in part by the inchworm method [49], which rearranges the bold hybridization diagram series to achieve improved performance for off-diagonal multi-orbital models [48]. However, applications have thus far been limited to Monte Carlo and quasi-Monte Carlo [15] implementations, with the accompanying low numerical accuracy. In the category of fast deterministic methods, tensor cross interpolation (TCI) and quantics TCI have been used for real time nonequilbrium steady state strong coupling diagrams [50] and imaginary time weak coupling diagrams in electron-phonon systems [51]. Related methods were earlier applied to bare hybridization expansions in real [52] and imaginary time [53]. The range of applicability of TCI-based methods is still being explored. Several works have used explicit expressions of the propagator along with symbolic computation, as in algorithmic Matsubara integration [54–57], or partial analytical evaluation combined with diagrammatic Monte Carlo [58], but these are limited to bare perturbation expansions. A more recent method of this type is more general, using the DLR of the propagator to obtain closed form Matsubara frequency sums which can be evaluated symbolically, reducing diagram evaluation to a sum over tensor contractions [59]. We lastly highlight the algorithm of Ref. 60, based on our approach, which uses Prony's method and SOE separation of variables to evaluate real time strong coupling diagrams for an electron-boson model up to third-order.

## II. BACKGROUND AND OVERVIEW

We refer to Ref. 43 for some background which is not repeated here. Specifically, Sec. IIIA-B describes the pseudo-particle strong coupling expansion of the Anderson impurity model and the accompanying diagrammatic notation, and Sec. IIID describes the DLR. We assume the reader is familiar with these topics. For comparison with our approach, it might also be useful to read Sec. IIIC on a naive "direct integration" algorithm for evaluating diagrams. To keep the present paper self-contained, we briefly review the diagrammatic notation so that mathematical expressions are clear. We then give a high-level overview of the idea of our algorithm.

We consider the Anderson impurity model with $n$ fermionic single-particle impurity states, so that the local many-body space has dimension $N = 2^n$. The hybridization function $\Delta(\tau)$ is an $n \times n$ matrix-valued function of imaginary time $\tau$. The creation operator for an impurity single-particle state $\kappa$ is denoted by $c_\kappa^\dagger$, and its matrix in a given local many-body basis is $F_{\kappa jk}^\dagger = \langle j|c_\kappa^\dagger|k\rangle$, where $|j\rangle$ is the $j$th local many-body basis state. We refer to the $N \times N$ matrix $F_\kappa^\dagger$ as a creation matrix. The pseudo-particle Green's function $\mathcal{G}(\tau)$ and self-energy $\Sigma(\tau)$ are $N \times N$ matrix-valued function satisfying a pseudo-particle Dyson equation (C5), which we solve in the DLR basis using techniques described in Ref. 44. Our goal is to compute the single-particle Green's function $G(\tau)$, an $n \times n$ matrix-valued function. The pseudo-particle strong coupling expansion yields diagrammatic expressions for the pseudo-particle self-energy $\Sigma$ and the single-particle Green's function $G$ in terms of $\Delta$ and $\mathcal{G}$. To obtain $G$ given $\Delta$, one must solve the pseudo-particle Dyson equation self-consistently to obtain $\mathcal{G}$, and then finally evaluate the diagrammatic expression for $G$.

We begin by describing the pseudo-particle self-energy diagrams, which are used as our primary example in the text. The single-particle Green's function diagrams are similar, and their structure is described in App. B. Diagrams of order $m \geq 1$ contain $m$ hybridization insertions, a combinatorially-growing number of diagram topologies, and $2^m$ combinations of propagation directions of the hybridization insertions. The number of diagram topologies at order $m$ is given by counting connected chord diagrams of $2m$ nodes [61, 62], and is denoted by $C(m)$. We use the notation $\Sigma_{j,k}^{(m)}(\tau)$ to refer to the pseudo-particle self-energy diagram of order $m$ with the $j$th topology and $k$th combination of hybridization di-

rections. Thus the pseudo-particle self-energy $\Sigma$ is given by $\Sigma = \sum_{m=1}^{\infty} \sum_{j=1}^{C(m)} \sum_{k=1}^{2^m} \Sigma_{j,k}^{(m)}$.

An illustrative example of the diagrammatic notation is given by the sole second-order (OCA) diagram:

$$
\begin{aligned}
\Sigma_{1,1}^{(2)}(\tau) =\ & \overset{\Delta_{\nu\lambda}\ \Delta_{\mu\kappa}}{\underset{\tau\quad \tau_2\quad \tau_1\quad 0}{\triangle\!\!-\!\!\triangle\!\!-\!\!\triangle\!\!-\!\!\triangle}} \\
=\ & c_{211} \int_0^\tau d\tau_2 \int_0^{\tau_2} d\tau_1\, \Delta_{\nu\lambda}(\tau - \tau_1)\Delta_{\mu\kappa}(\tau_2) \\
& \times F_\nu^\dagger\, \mathcal{G}(\tau - \tau_2)\, F_\mu^\dagger\, \mathcal{G}(\tau_2 - \tau_1)\, F_\lambda\, \mathcal{G}(\tau_1)\, F_\kappa.
\end{aligned}
\tag{1}
$$

Here and in the rest of the text, there is an implied sum over the impurity state indices $\kappa, \lambda, \mu, \nu = 1, \ldots, n$. The prefactor $c_{mjk}$ is $\pm 1$, depending on the diagram order, topology, and hybridization directions. An $m$th-order diagram is a $2m-2$-dimensional integral-valued function of $\tau$. The diagram is composed first of a "backbone" line, which specifies the external time variable $\tau$, a series of internal time variables $\tau_1, \ldots, \tau_{2m-2}$, and their order of integration. Each backbone line implies an insertion of a pseudo-particle Green's function $\mathcal{G}$, with argument determined by its two vertices. There are then $m$ hybridization lines, implying an insertion of a hybridization function $\Delta$ and a pair of creation ($F^\dagger$) and annihilation ($F$) matrices, with the arguments of $\Delta$ and the locations of the creation/annihilation matrices determined by the two vertices of the hybridization line. Creation matrices are indicated by green triangles, and annihilation matrices by red triangles. The arrow, or direction, of the hybridization line indicates the sign of the argument of the hybridization insertion (for the leftmost insertion above, $\Delta_{\nu\lambda}(\tau - \tau_1)$ for a "forward" line and $\Delta_{\lambda\nu}(\tau_1 - \tau)$ for a "backward" line), and the order of the creation/annihilation matrix insertions ($F_\nu^\dagger$ on the left vertex, $F_\lambda$ on the right vertex for forward lines, and $F_\lambda$ on the left vertex, $F_\nu^\dagger$ on the right vertex for backward lines). As another example, we have

$$
\begin{aligned}
\Sigma_{1,2}^{(2)}(\tau) =\ & \overset{\Delta_{\nu\lambda}\ \Delta_{\kappa\mu}}{\underset{\tau\quad \tau_2\quad \tau_1\quad 0}{\triangle\!\!-\!\!\triangle\!\!-\!\!\triangle\!\!-\!\!\triangle}} \\
=\ & c_{212} \int_0^\tau d\tau_2 \int_0^{\tau_2} d\tau_1\, \Delta_{\nu\lambda}(\tau - \tau_1)\Delta_{\kappa\mu}(-\tau_2) \\
& \times F_\nu^\dagger\, \mathcal{G}(\tau - \tau_2)\, F_\kappa\, \mathcal{G}(\tau_2 - \tau_1)\, F_\lambda\, \mathcal{G}(\tau_1)\, F_\mu^\dagger.
\end{aligned}
\tag{2}
$$

For fixed impurity state indices $\kappa, \lambda, \mu, \nu$ and fixed time arguments, the integrand is given by a sequence of $N \times N$ matrix-matrix multiplications between the pseudo-particle Green's functions and the creation/annihilation matrices, followed by scalar multiplications by the hybridization functions. There is a convolutional structure in time between the pseudo-particle Green's functions, which is broken by some of the hybridization insertions, resulting in a full $2m-2$-dimensional integral.

**Remark 1.** *To avoid confusion, we highlight a difference between the diagrammatic notation used here and that in Ref. 43: here, the impurity state variables appearing in a diagrammatic expression (e.g., $\kappa, \lambda, \mu, \nu$ in (1)) are implicitly considered to be summed over, while in Ref. 43, they were taken to be fixed, with the sum taken later. Indeed, for the present algorithm, we have observed that it is more efficient to consider this full sum at once, rather than to evaluate each expression with fixed impurity state indices individually and then perform the sum after as was done in Ref. 43.*

The basic idea of our algorithm is as follows. We define the kernel

$$
K(\tau, \omega) = -\frac{e^{-\omega\tau}}{1 + e^{-\beta\omega}},
\tag{3}
$$

and will represent the hybridization function as a $p$-term expansion

$$
\Delta_{\nu\lambda}(\tau) \approx \sum_{l=1}^p \Delta_{\nu\lambda l} K(\tau, \omega_l).
\tag{4}
$$

This is an SOE approximation. Substituting this expression for the hybridization function $\Delta_{\nu\lambda}(\tau - \tau_1)$ in (1), for example, we can use the exponential sum rule $e^{a+b} = e^a e^b$ to separate the $\tau$ and $\tau_1$ variables (ignoring overflow issues, which will be handled later). We are left with a sum over $p$ terms, each of which can be rearranged as a sequence of products and one-dimensional convolutions. We will also show that some of the sums over impurity state indices (in this case, $\nu$) can be precomputed to simplify the resulting expression. Finally, we use the DLR to perform all imaginary time products and convolutions efficiently. This will yield an evaluation algorithm with computational complexity

$$
\mathcal{O}((np)^{m-1}(mr^2N^3 + nrN^3)).
\tag{5}
$$

Here $r$ is the number of degrees of freedom in the DLR, which scales mildly as $r = \mathcal{O}(\log(\Lambda)\log(\epsilon^{-1}))$, where $\Lambda = \beta\omega_{\max}$ is the dimensionless product of the inverse temperature and an upper bound on the spectral width of all quantities, and $\epsilon$ is the desired accuracy. This improves the $\mathcal{O}(mn^{2m}r^{m+1}N^3)$ complexity of the algorithm presented in Ref. 43, particularly because we find a fit with $p < r$ can always be achieved in practice.

Sec. III will present an algorithm to obtain a compact SOE approximation (4), with illustrative examples presented for several example hybridization functions. Sec. IV will describe the diagram evaluation algorithm in detail. Finally, Sec. V will present several benchmark examples, including timing results.

## III. HYBRIDIZATION FITTING USING THE AAA ALGORITHM

The factor $p^{m-1}$ appearing in the computational complexity (5) is a consequence of the $m$ hybridization functions in an $m$th-order diagram (separation of variables

will only need to be applied to $m - 1$ of them). It will therefore be crucial to obtain an approximation with $p$ as small as possible. In Ref. 43, such an expansion was obtained using the discrete Lehmann representation (DLR) [44, 63]. The DLR provides a universal collection of frequencies $\omega_l$ such that an expansion of the form (4) holds to a controllable accuracy for any imaginary time hybridization function with a given spectral width and temperature. The DLR frequencies $\omega_l$ depend only on the dimensionless parameter $\Lambda = \beta\omega_{\max}$, where $\beta$ is the inverse temperature and $\omega_{\max}$ is the spectral width, and on the desired accuracy $\epsilon$. They do not depend on the specific structure of $\Delta_{\nu\lambda}(\tau)$. This is made possible by the numerical low-rank of the analytic continuation operator: quantities which have been analytically continued to the imaginary axis occupy a subspace whose dimension, to within $\epsilon$ accuracy, scales mildly as $p = \mathcal{O}(\log(\Lambda)\log(\epsilon^{-1}))$, superior to generic high-order accurate representations such as orthogonal polynomials [64–66]. These properties make the DLR (and the closely related intermediate representation [67–69]) useful as a standard discretization for imaginary time quantities, such as in various operations in the DMFT loop including the solution of the Dyson equation [63, 70–72], discretization of the imaginary time branch of the Keldysh contour [73, 74], and compact representations of two-particle quantities [75–77]. For further details on the DLR, we refer to Refs. 44 and 63, or to the short description in Ref. 43.

Since our goal is to obtain an SOE approximation for a fixed hybridization function $\Delta_{\nu\lambda}(\tau)$, the universality of the DLR might cause it to be suboptimal. Indeed, in this case it is preferable to tailor the frequencies $\omega_l$ to $\Delta_{\nu\lambda}(\tau)$ itself. For instance, if the spectral density of $\Delta(\tau)$ is given by $\rho(\omega) = \delta(\omega - \omega_0)$, then $\Delta(\tau) = K(\tau, \omega_0)$, and $\Delta$ can be represented exactly with $p = 1$. More specifically, the universality of the DLR implies that the number of DLR basis functions for a given $\Lambda$ and $\epsilon$ is an upper bound on the optimal $p$.

In general, finding an optimal SOE approximation of a given hybridization function is a challenging nonlinear optimization problem, but we will present an efficient algorithm based on the AAA algorithm for rational approximation [45] and nonlinear optimization which yields more compact expansions than the DLR. We note that our algorithm is closely related to that of Ref. 46 for the analytic continuation problem, but with some modifications (for example, we explicitly enforce an approximation by simple poles on the real axis, as described in Sec. III B). We also point out that similar hybridization fitting methods have found recent use in quantum many-body applications outside of diagrammatics, particularly at low temperatures, for example in pseudomode theories for non-Markovian quantum systems [78, 79], and the hierarchichal equations of motion approach [80].

We begin by Fourier transforming (4) to obtain

$$\Delta_{\nu\lambda}(i\nu_n) \approx \sum_{l=1}^{p} \Delta_{\nu\lambda l} K(i\nu_n, \omega_l) = \sum_{l=1}^{p} \frac{\Delta_{\nu\lambda l}}{i\nu_n - \omega_l}, \quad (6)$$

where we have defined the fermionic analytic continuation kernel in Matsubara frequency as

$$K(i\nu_n, \omega) = \int_0^{\beta} d\tau\, e^{i\nu_n\tau} K(\tau, \omega) = \frac{1}{i\nu_n - \omega}, \quad (7)$$

with $i\nu_n = (2n + 1)\pi i/\beta$. We thus have a rational approximation problem: determine simple pole locations $\omega_l$ and corresponding residues $\Delta_{\nu\lambda}$ such that (6) holds to a user-specified accuracy $\epsilon$, with $p$ as small as possible.

## A. Overview of AAA algorithm

The adaptive Antoulas–Anderson (AAA) algorithm [45] is a method of approximating a function $f : D \to \mathbb{C}$, with $D \subset \mathbb{C}$ some domain (we will take $D = \{i\nu_n\}_{n=-\infty}^{\infty}$), by a rational function. It constructs an interpolant of $f$ in the following barycentric form:

$$r^{(k)}(z) = \frac{n(z)}{d(z)} = \sum_{j=1}^{k} \frac{w_j f_j}{z - z_j} \Big/ \sum_{j=1}^{m} \frac{w_j}{z - z_j}. \quad (8)$$

Here $z_j \in D$ are called support points, $f_j = f(z_j)$, $w_j \in \mathbb{C}$ are called weights, and $n(z)$, $r(z)$ refer to the numerator and denomator as written, themselves rational functions. It is straightforward to verify that as long as $w_j \neq 0$, $\lim_{z \to z_j} r(z) = f_j$, i.e., $z_j$ is a removable singularity and $r(z)$ interpolates $f(z)$ at the support points. The AAA algorithm builds such an approximant from values of $f$ at a collection of $K$ sample points $Z = \{Z_1, \ldots, Z_K\} \subset D$.

The algorithm adds support points $z_j$ from the available sample points iteratively, and computes corresponding weights $w_j$. Suppose an approximant $r^{(k-1)}$ has already been constructed (if $k = 1$, we begin with $r^{(0)} = 0$) with support points $\mathcal{Z}^{(k-1)} = \{z_1, \ldots, z_{k-1}\}$. The point $z_k \in Z$ at which the error of $r^{(k-1)}$ is maximal is selected:

$$z_k = \underset{z \in Z \backslash \mathcal{Z}^{(k-1)}}{\arg\max} |r^{(k-1)}(z) - f(z)|. \quad (9)$$

This point is removed from the collection of sample points, and added to the collection of support points. The weights $w_1, \ldots, w_{k-1}$ are then recomputed, along with the new weight $w_k$, in order to minimize the residual of the approximant at the remaining sample points, in the following sense:

$$w^{(k)} = \underset{\|w^{(k)}\|_2 = 1}{\arg\min} \sum_{z \in Z \backslash \mathcal{Z}^{(k)}} |d(z)f(z) - n(z)|^2. \quad (10)$$

Here $w^{(k)} = (w_1, \ldots, w_k)$, and $\|\cdot\|_2$ denotes the Euclidean norm. Since $d(z)f(z) - n(z) = \sum_{j=1}^{k} \frac{f(z) - f_j}{z - z_j} w_j$, (10) is

equivalent to the following problem

$$w^{(k)} = \arg\min_{\|w^{(k)}\|_2 = 1} \left\| A w^{(k)} \right\|_2, \tag{11}$$

where $A$ is the $(K - k) \times k$ matrix given by

$$A_{ij} = \frac{f(\zeta_i) - f_j}{\zeta_i - z_j}, \quad \zeta_i \in Z \backslash \mathcal{Z}^{(k)}. \tag{12}$$

The solution is the right singular vector of $A$ corresponding to its smallest singular value, and can be computed by an SVD. This yields the interpolant $r^{(k)}$, and the process can be repeated to further improve its accuracy.

In our application, it will be necessary to obtain the rational interpolant $r^{(k)}(z)$ as a sum of simple poles, rather than in the barycentric form (8). The pole locations are given by the zeros of $d(z)$, and it can be shown that these are the finite eigenvalues of the following generalized eigenvalue problem [45]:

$$\begin{pmatrix} 0 & w_1 & w_2 & \cdots & w_k \\ 1 & z_1 & & & \\ 1 & & z_2 & & \\ \vdots & & & \ddots & \\ 1 & & & & z_k \end{pmatrix} v = \lambda \begin{pmatrix} 0 & & & & \\ & 1 & & & \\ & & 1 & & \\ & & & \ddots & \\ & & & & 1 \end{pmatrix} v. \tag{13}$$

We assume here that the poles of $r^{(k)}$ are in fact simple, and that $n(z)$ and $d(z)$ do not have common roots; we have observed that this is the case in practice. We note that we do not require the pole residues, as the final hybridization expansion coefficients will be determined within a subsequent optimization procedure described in Sec. III C.

### B. Application to hybridization fitting

Taking $D = \{i\nu_n\}_{n=-\infty}^{\infty}$ and $f = \Delta$, the AAA algorithm produces a rational approximation for the hybridization function in the case that it is scalar-valued. In this work, we require the following two simple modifications to the AAA algorithm: (1) We must generalize the AAA algorithm to handle the matrix-valued function $\Delta_{\nu\lambda}(i\nu_n)$, and (2) We must enforce that the simple poles obtained by the AAA algorithm fall on the real axis. The reason for the second point will become clear when we discuss our separation of variables procedure—complex frequencies $\omega_l$ in the expansion (4) would lead to oscillating exponentials which are not well-represented in the DLR basis used for efficient convolution.

To address the first point, we treat $\Delta_{\nu\lambda}(i\nu_k)$ as a vector-valued function (indexed over $\nu$ and $\lambda$), and consider the generalization of the AAA algorithm to vector-valued $f$. This simply requires replacing the absolute values in (9) and (10) by the Euclidean norm. The minimization problem (10) can still be solved using the SVD, but with the row dimension multiplied by the dimension of $f$. We note that a matrix-valued AAA algorithm

has been proposed in which the weights $w_j$ are taken to be matrix-valued [81], but we have found the simpler vector-valued scheme to be adequate and effective for our purposes.

We address the second point by enforcing the symmetry condition $\Delta(-i\nu_n) = \Delta(i\nu_n)^\dagger$ on the rational interpolant. This property can be seen from the Lehmann representation

$$\Delta_{\nu\lambda}(i\nu_n) = \int_{-\infty}^{\infty} d\omega \, \frac{\rho_{\nu\lambda}(\omega)}{i\nu_n - \omega}, \tag{14}$$

where $\rho$ is a Hermitian spectral density. This representation furthermore shows that the Matsubara data is a weighted combination of simple poles on the real axis, and is therefore inconsistent with a simple pole approximation containing poles off the real axis. Although we have not proven that the symmetrization condition is sufficient to guarantee real pole locations (in principle, complex conjugate pairs of poles still seem to be allowed), we observe in practice that this property, as well as the exact inconsistency of complex-valued poles with the Matsubara data, is sufficient to produce real pole locations to within machine precision. The details of our symmetrized AAA procedure are given in App. A.

### C. Refinement of pole approximation via bilevel optimization

Although our modified AAA algorithm can already be used to produce an efficient greedy solution of the hybridization fitting problem, the approximation with a given number of poles can be made more accurate by adjusting the pole locations using the bilevel optimization procedure described in Refs. 46 and 47. We define the fitting error

$$\text{Err}\left(\{\omega_l, \Delta_l\}_{l=1}^p\right) = \sum_{n=-\infty}^{\infty} \left\| \Delta(i\nu_n) - \sum_{l=1}^p \frac{\Delta_l}{i\nu_n - \omega_l} \right\|_F^2, \tag{15}$$

where $\Delta_l$ is the matrix with entries $\Delta_{\nu\lambda l}$, and $\|\cdot\|_F$ is the Frobenius norm. We then define an objective function

$$\mathcal{E}(\{\omega_l\}_{l=1}^p) = \min_{\{\Delta_l\}_{l=1}^p} \text{Err}\left(\{\omega_l, \Delta_l\}_{l=1}^p\right). \tag{16}$$

Given pole locations $\{\omega_l\}_{l=1}^p$, $\mathcal{E}$ can be straightforwardly computed by solving an overdetermined least squares problem (with the Matsubara frequency sum in (15) suitably truncated). Minimization of $\mathcal{E}$ with respect to $\{\omega_l\}_{l=1}^p$ therefore defines a bilevel optimization problem, which we solve using the L-BFGS algorithm [82], taking the pole locations produced by the modified AAA procedure run for $p/2$ iterates (to produce a sum of $p$ poles) as an initial guess. We refer to Ref. 46 for further details on this algorithm.

Our full procedure to obtain an SOE approximation (4) is as follows. We first specify the desired number of

poles $p$, which is an even integer. We then run the AAA procedure to obtain an approximation by $p$ symmetrized poles, and use the bilevel optimization procedure to adjust the pole locations and residues, minimizing the approximation error (15). We then measure the error of the resulting SOE approximation (4), obtained via analytical Fourier transform of (6), with respect to the Frobenius $L^2$ norm given by

$$\left( \sum_{\nu,\lambda=1}^{n} \frac{1}{\beta} \int_0^\beta d\tau \, |f_{\nu\lambda}(\tau)|^2 \right)^{1/2}. \qquad (17)$$

If the error is larger than a desired tolerance $\epsilon$, we increase $p$ and repeat the procedure.

### D.  Numerical examples

In Fig. 1, we demonstrate the performance of our hybridization fitting algorithm for several inverse temperatures $\beta$, using hybridization functions given by (14) with the following spectral densities:

- Sum of $\delta$-functions: $\rho(\omega) = \sum_{k=1}^{3} c_k \delta(\omega - \omega_k)$

- Semicircle: $\rho(\omega) = \frac{2}{\pi}\sqrt{1-\omega^2}$

- Sum of Gaussians: $\rho(\omega) = \sum_{k=1}^{3} c_k e^{-(a_k(\omega-\omega_k))^2}$.

For the sum of $\delta$-functions example, we take $\omega_{\{1,2,3\}} = \{-0.3, -0.1, 0.2\}$ and $c_{\{1,2,3\}} = \{0.2, 0.55, 0.25\}$. For the sum of Gaussians example, we take $\omega_{\{1,2,3\}} = \{0.5, 0.08, -0.7\}$, $c_{\{1,2,3\}} = \{0.2, 0.5, 0.3\}$, and $a_{\{1,2,3\}} = \{20, 30, 15\}$. Accurate reference solutions can be computed analytically or by high-order adaptive quadrature. We observe that for the sum of $\delta$-functions example, our algorithm recovers the hybridization to within nearly machine precision using only three poles, regardless of $\beta$. For the continuous spectra, the fitting error decreases approximately exponentially with the number of poles, at a decreasing rate with increasing $\beta$.

We study the dependence on $\beta$ more closely in Fig. 2, measuring the minimum number of poles required to achieve an accuracy $\epsilon = 10^{-6}$. For the two examples of continuous spectra (left panel), we observe a logarithmic dependence of the number of poles $p$ on $\beta$. This data is consistent with the scaling $p = \mathcal{O}(\log(\Lambda)\log(\epsilon^{-1}))$ achieved by the DLR (recall $\Lambda = \beta\omega_{\max}$). However, by also plotting the number $r$ of DLR basis functions for given $\beta$ and $\epsilon$, we observe that the prefactor is smaller by a factor roughly two for the semicircle case and four for the sum of Gaussians case. This is not surprising, since the DLR basis at fixed $\Lambda = \beta\omega_{\max}$ and $\epsilon$ is sufficient to represent *all* functions of the form (14) to within $\epsilon$ accuracy, whereas our fitting algorithm is tailored to a *specific* given hybridization function. The reduction is imporant in practice because $p$ enters as the base of an exponential scaling with respect to diagram order in the

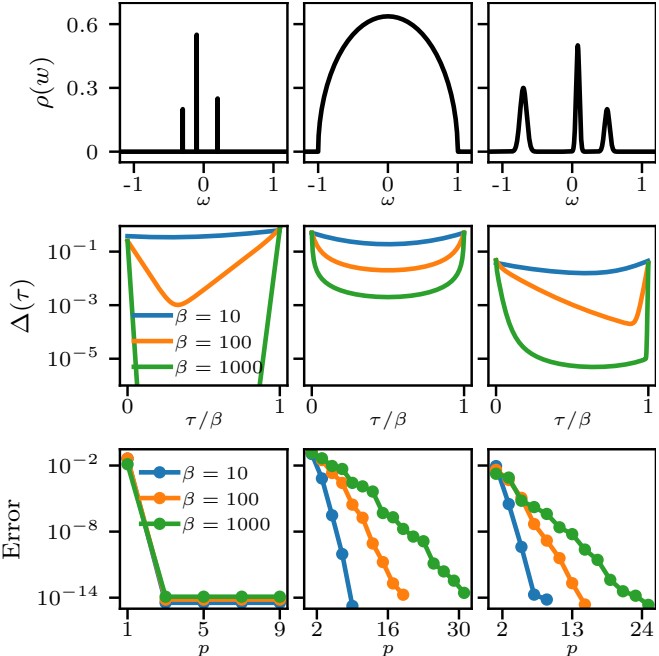

FIG. 1. Fitting of hybridization functions (second row) generated from three spectral densities (first row) for several inverse temperatures. Fitting errors (17) versus the number of poles $p$ are shown in the third row. We observe essentially exact fitting for discrete spectra, and systematic convergence for continuous spectra.

computational complexity (5) of our diagram evaluation algorithm.

We further explore the expected reduction by considering a "worst case scenario" random pole model,

$$\rho(\omega) = \sum_{j=1}^{N} \delta(\omega - \omega_j) v_j v_j^\dagger, \qquad (18)$$

with $\omega_j$ uniformly randomly sampled on $[-1, 1]$ and $v_j \in \mathbb{C}^M$ a random vector such that $\sum_{j=1}^{N} \|v_j\|_2^2 = 1$. We can then generate Matsubara data using (14), and apply our hybridization fitting algorithm with an error tolerance $\epsilon = 10^{-6}$. We consider two cases: single-orbital random discrete spectrum ($M = 1$), and multi-orbital ($M = N$). In Fig. 2, for both cases, we plot the required number of poles $p$ against the actual number of poles $N$ in the spectrum, for 1200 random experiments at $\beta = 100$. We also plot the number of DLR basis functions with the same error tolerance for comparison. We observe that in both cases, we find $p = N$ until a saturation point which is approximately half the number of DLR basis functions for $\beta = 100$. In the left panel, we plot this saturation point for the multiorbital case against $\beta$ ("Random poles" label). This experiment therefore provides further evidence that our fitting algorithm outperforms the DLR for any fixed hybridization function.

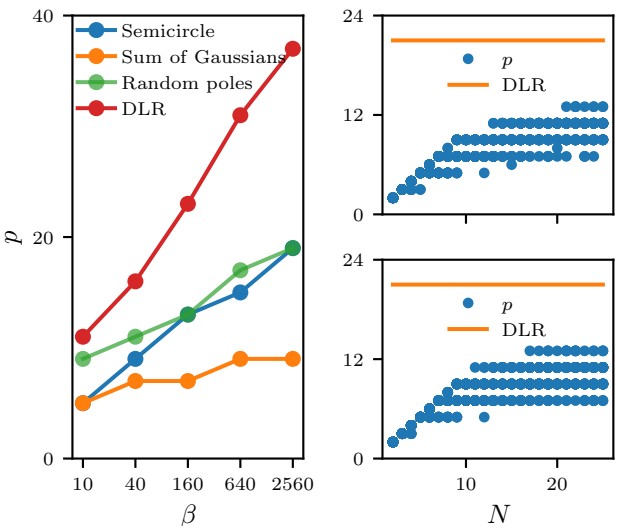

FIG. 2. (Left) Number of poles $p$ required by our hybridization fitting algorithm with tolerance $10^{-6}$ vs. inverse temperature $\beta$, for semicircle and sum of Gaussians spectra, a many-orbital random pole model ($p$ saturates to the plotted value as the number of random orbitals is increased; see bottom-right panel), as well as the number of DLR basis functions for equivalent parameters. (Right) Number of poles $p$ required to achieve accuracy $10^{-6}$ vs. actual number of poles $N$ in a random pole model, for single-orbital (top) and multi-orbital (bottom) cases with $\beta = 100$ (data for 1200 random experiments each).

## IV. DECOMPOSING DIAGRAMS VIA SUM-OF-EXPONENTIALS APPROXIMATION

Given the SOE approximation (4) of the hybridization function, we can formulate our diagram evaluation algo-

rithm. We will use the properties

$$K(\tau - \tau', \omega) = \frac{K(\tau, \omega)K(\tau', -\omega)}{K(0, -\omega)} \tag{19}$$

$$K(\tau - \tau', \omega) = \frac{K(\tau - \tau'', \omega)K(\tau'' - \tau', \omega)}{K(0, \omega)} \tag{20}$$

of the kernel (3), where $\tau''$ is an auxiliary imaginary time variable whose selection will be discussed below. These allow us to separate variables in (4):

$$\Delta_{\nu\lambda}(\tau - \tau') \approx \sum_{\omega_l \leq 0}^{p} \frac{\Delta_{\nu\lambda l}}{K_l^-(0)} K_l^+(\tau) K_l^-(\tau')$$
$$+ \sum_{\omega_l > 0}^{p} \frac{\Delta_{\nu\lambda l}}{K_l^+(0)} K_l^+(\tau - \tau'') K_l^+(\tau'' - \tau'). \tag{21}$$

We have introduced the shorthand $K_l^\pm(\tau) = K(\tau, \pm\omega_l)$. We split the sums into $\omega_l \leq 0$ and $\omega_l > 0$ terms because $\frac{1}{K^-(0)} = 1 + e^{\beta\omega}$, so an expansion using (19) alone is vulnerable to numerical overflow. If $0 \leq \tau' \leq \tau'' \leq \tau \leq \beta$, then $\tau - \tau'', \tau'' - \tau' \in [0, \beta]$, and the use of (20) for terms with $\omega_l > 0$ avoids this issue.

### A. One-crossing approximation self-energy

As an example of our procedure, we consider the OCA diagram (1). If the hybridization line $\Delta_{\nu\lambda}(\tau - \tau_1)$ were removed, then (i) the imaginary time integrals could be computed as a sequence of two nested convolutions, rather than a double integral, and (ii) certain sums over impurity state indices could be moved inside the expression and performed in advance. Our decomposition procedure uses separation of variables to obtain a sum over simplified diagrams for which this holds true.

Substituting in (21) with $\tau' = \tau_1$ and $\tau'' = \tau_2$ (note $\tau_1 \leq \tau_2 \leq \tau$) into (1) yields (note that $c_{mjk}^2 = 1$)

$$c_{211} \; \overset{\Delta_{\nu\lambda} \; \Delta_{\mu\kappa}}{\underset{\tau \quad \tau_2 \quad \tau_1 \quad 0}{\triangle \; \triangle \; \triangle \; \triangle}} = \sum_{\omega_l \leq 0} \frac{K_l^+(\tau)}{K_l^-(0)} \overline{F}_{\lambda l}^\dagger \int_0^\tau d\tau_2 \, \mathcal{G}(\tau - \tau_2) F_\mu^\dagger \Delta_{\mu\kappa}(\tau_2) \int_0^{\tau_2} d\tau_1 \, \mathcal{G}(\tau_2 - \tau_1) F_\lambda (\mathcal{G}K_l^-)(\tau_1) F_\kappa$$

$$+ \sum_{\omega_l > 0} \frac{1}{K_l^+(0)} \overline{F}_{\lambda l}^\dagger \int_0^\tau d\tau_2 \, (\mathcal{G}K_l^+)(\tau - \tau_2) F_\mu^\dagger \Delta_{\mu\kappa}(\tau_2) \int_0^{\tau_2} d\tau_1 \, (\mathcal{G}K_l^+)(\tau_2 - \tau_1) F_\lambda \mathcal{G}(\tau_1) F_\kappa$$

$$= \sum_{\omega_l \leq 0} \frac{1}{K_l^-(0)} \underset{\tau \quad \mathcal{G} \quad \tau_2 \quad \mathcal{G} \quad \tau_1 \quad 0}{\overset{K_l^+ \overline{F}_{\lambda l}^\dagger \quad F_\mu^\dagger \Delta_{\mu\kappa} \quad F_\lambda \mathcal{G}K_l^- \quad F_\kappa}{\longleftarrow}} + \sum_{\omega_l > 0} \frac{1}{K_l^+(0)} \underset{\tau \quad \mathcal{G}K_l^+ \quad \tau_2 \quad \mathcal{G}K_l^+ \quad \tau_1 \quad 0}{\overset{\overline{F}_{\lambda l}^\dagger \quad F_\mu^\dagger \Delta_{\mu\kappa} \quad F_\lambda \mathcal{G} \quad F_\kappa}{\longleftarrow}}. \tag{22}$$

Here, we have defined

$$\overline{F}_{\lambda l}^\dagger = \sum_{\nu=1}^n \Delta_{\nu\lambda l} F_\nu^\dagger \tag{23}$$

and introduced the notation $(\mathcal{G}K_l^\pm)(\tau) = \mathcal{G}(\tau)K_l^\pm(\tau)$. We note that precomputing $\overline{F}^\dagger$ leads to an $\mathcal{O}(n^m)$ reduction in computational complexity compared to the

algorithm proposed in Ref. 43. In (22), and in the remainder of the text, the sums over $\omega_l$ are written explicitly to emphasize the partitioning of the poles into positive and negative frequencies, but we remind the reader that the remaining sums over orbital indices are understood implicitly. We have also introduced a diagrammatic notation for the integral expressions. Each term is represented by a backbone diagram with vertices at 0, $\tau_1$, $\tau_2$, and $\tau$. These backbone diagrams are evaluated from right to left, with edges correspond to time-ordered imaginary time convolution, and vertices corresponding to imaginary time multiplication.

In this way, we have decomposed the diagram into a sum over sequences of imaginary time products and convolutions, as well as matrix-matrix products in the pseudo-particle Hilbert space, eliminating the double integral over imaginary time variables. Let us consider the evaluation of (22). We use the DLR [44, 63] to discretize all imaginary time quantities on a grid of $r = \mathcal{O}(\log(\Lambda)\log(\epsilon^{-1}))$ imaginary time nodes. Using the DLR, a product of imaginary time functions can be computed in $\mathcal{O}(r)$ operations (by multiplying their values on the DLR grid), and a convolution in $\mathcal{O}(r^2)$ operations (see Ref. 43, App. A). We perform the sums over $l$ and $\lambda$ explicitly. For fixed $\omega_l \leq 0$ and $\lambda$, we can evaluate the corresponding backbone diagram as follows, with the complexity of each step indicated in parentheses:

1. Compute $F_\lambda \mathcal{G}(\tau_1)K_l^-(\tau_1)$ $(\mathcal{O}(rN^3))$.

2. Convolve by $\mathcal{G}$ $(\mathcal{O}(r^2N^3))$.

3. For each $\kappa$, multiply by $F_\kappa$ from the right $(\mathcal{O}(nrN^3))$.

4. For each $\mu, \kappa$, multiply by $\Delta_{\mu\kappa}$, and sum over $\kappa$ $(\mathcal{O}(n^2rN^2))$.

5. Multiply by $F_\mu^\dagger$ and sum over $\mu$ $(\mathcal{O}(nrN^3))$.

6. Convolve by $\mathcal{G}$ $(\mathcal{O}(r^2N^3))$.

7. Multiply by $K_l^+ \overline{F}_{\lambda l}^\dagger$ $(\mathcal{O}(rN^3))$.

The $\omega_l > 0$ sum can be evaluated similarly. Including the sum over $l$, and $\lambda$ and using that $N > n$, we obtain a total computational complexity

$$\mathcal{O}(np\,(r^2N^3 + nrN^3)). \tag{24}$$

**Remark 2.** *The order of operations described above has a smaller computational complexity than a simple "right-to-left" evaluation of the diagram, which scales as $\mathcal{O}(n^2pr^2N^3)$. The improvement comes from waiting to multiply by the right-most annihilation matrix $F_\kappa$ until just before multiplying by the hybridization $\Delta_{\lambda\kappa}$ which connects to $\tau = 0$.*

**Remark 3.** *Since $K(\tau' - \tau, \omega) = -K(\tau - \tau', -\omega)$ for $\tau > \tau'$, we have*

$$\Delta_{\lambda\nu}(\tau' - \tau) \approx -\sum_{\omega_l \leq 0}^{p} \frac{\Delta_{\lambda\nu l}}{K_l^+(0)} K_l^-(\tau)K_l^+(\tau')$$
$$-\sum_{\omega_l > 0}^{p} \frac{\Delta_{\lambda\nu l}}{K_l^-(0)} K_l^-(\tau - \tau'')K_l^-(\tau'' - \tau'). \tag{25}$$

*We also define*

$$\overline{F}_{\nu l} = \sum_{\lambda=1}^{n} \Delta_{\lambda\nu l}F_\lambda. \tag{26}$$

*Thus, a backward propagation line can be handled similarly to a forward line by swapping $K_l^+$ with $K_l^-$, $\overline{F}_{\lambda l}^\dagger$ with $\overline{F}_{\nu l}$, $F_\lambda$ with $F_\nu^\dagger$, and multiplying by $-1$.*

## B. Decomposing general diagrams

Decomposing the OCA self-energy diagram (1) using our procedure only requires separating variables for a single hybridization line. To understand the details of the procedure for general diagrams, it is useful to consider a diagram for which at least two hybridization lines must be separated, which occurs at third-order and higher. As a concrete example, we consider the following third-order pseudo-particle self-energy diagram:

$$\Sigma_{1,1}^{(3)}(\tau) = \overset{\Delta_{\pi\nu} \quad \Delta_{\xi\lambda} \quad \Delta_{\mu\kappa}}{\underset{\tau \quad \tau_4 \quad \tau_3 \quad \tau_2 \quad \tau_1 \quad 0}{\triangle}}. \tag{27}$$

Here, the hybridization lines $\Delta_{\pi\nu}(\tau - \tau_3)$ and $\Delta_{\xi\lambda}(\tau_4 - \tau_1)$ break the convolutional structure of the backbone, and must be separated in the manner described above. However, we encounter a problem: it is unclear how to choose $\tau''$ in (21) for $\Delta_{\xi\lambda}(\tau_4 - \tau_1)$ in a way that maintains the convolutional structure. To address this, we generalize (20) as follows:

$$K(\tau - \tau', \omega)$$
$$= \frac{K(\tau - \tau^{(1)}, \omega)K(\tau^{(1)} - \tau^{(2)}, \omega)\cdots K(\tau^{(j)} - \tau', \omega)}{K^j(0, \omega)}. \tag{28}$$

This leads to the decomposition

$$\Delta_{\xi\lambda}(\tau_4 - \tau_1) \approx \sum_{\omega_l \leq 0}^{p} \frac{\Delta_{\xi\lambda l}}{K_l^-(0)} K_l^+(\tau_4)K_l^-(\tau_1)$$
$$+ \sum_{\omega_l > 0}^{p} \frac{\Delta_{\xi\lambda l}}{(K_l^+(0))^2} K_l^+(\tau_4 - \tau_3)$$
$$\times K_l^+(\tau_3 - \tau_2)K_l^+(\tau_2 - \tau_1). \tag{29}$$

For backward propagation, we follow Remark 3, but use

$$K(\tau' - \tau, \omega) = \frac{(-1)^j}{K^j(0, -\omega)} K(\tau' - \tau^{(j)}, -\omega)$$
$$\times K(\tau^{(j)} - \tau^{(j-1)}, -\omega) \cdots K(\tau^{(1)} - \tau, -\omega).$$

(30)

We now proceed in steps, beginning from (27):

$$
\begin{aligned}
c_{311} \Sigma^{(3)}_{1,1}(\tau) = \quad &\text{[backbone diagram]} \\[4pt]
= \sum_{\omega_l \le 0} \frac{1}{K_l^-(0)} &\text{[diagram]} + \sum_{\omega_l > 0} \frac{1}{K_l^+(0)} \text{[diagram]} \\[4pt]
= \sum_{\omega_l \le 0, \omega_{l'} \le 0} \frac{1}{K_l^-(0) K_{l'}^-(0)} &\text{[diagram]} \\[4pt]
+ \sum_{\omega_l \le 0, \omega_{l'} > 0} \frac{1}{K_l^-(0)(K_{l'}^+(0))^2} &\text{[diagram]} \\[4pt]
+ \sum_{\omega_l > 0, \omega_{l'} \le 0} \frac{1}{K_l^+(0) K_{l'}^-(0)} &\text{[diagram]} \\[4pt]
+ \sum_{\omega_l > 0, \omega_{l'} > 0} \frac{1}{K_l^+(0)(K_{l'}^+(0))^2} &\text{[diagram]} .
\end{aligned}
$$

(31)

Here, we have written out the decomposition procedure in our diagrammatic notation, which can be translated into integral formulas as in (22). In the first equality, we notate the action of the hybridization line $\Delta_{\mu\kappa}$ connecting $\tau_2$ to 0 (which appears as a multiplication) and the backbone propagators $\mathcal{G}$ in the same manner as in (22) for the OCA self-energy diagram. In the second, we separate variables in the hybridization line $\Delta_{\pi\nu}$ using the expansion (21), also in the same manner as in (22) (with $\tau' = \tau_3$ and $\tau'' = \tau_4$). In the third, we decompose $\Delta_{\xi\nu}$ using (29), yielding four sums of backbone diagrams.

This example can be generalized to a systematic procedure to decompose diagrams of any order and topology. We state this procedure in terms of our diagrammatic notation, but the steps simply encode the decomposition of hybridization functions via (29), and the rearrangement of the resulting nested sums and integrals. Subsequently, the diagrammatic notation can be translated into integral

expressions involving imaginary time products, convolutions, and matrix-matrix products in the pseudo-particle Hilbert space. We imagine that each vertex and edge of a backbone diagram is labeled with a function $f$, with an empty label indicating the constant function 1, and that "placing" a function $g$ on that vertex or edge means replacing $f$ by the product $fg$. In the resulting backbone diagrams, edges correspond to imaginary time convolution by their associated function, and vertices to multiplication. We state each step for forward propagating hybridization lines, and indicate the alternative procedure for backward propagating lines in parentheses. To understand the steps, we recommend following the example (31) for the self-energy diagram (27).

1. Precompute $\overline{F}^\dagger$ as in (23) ($\overline{F}$ as in (26)).

2. Beginning with a diagram containing $m$ hybridization lines, label each backbone line by the prop-

agator $\mathcal{G}$, except for the line connecting $\tau_1$ to 0, for which the propagator is placed on the $\tau_1$ vertex (since it represents a multiplication, rather than a convolution).

3. Eliminate the hybridization line $\Delta_{\mu\kappa}(\tau)$ $(\Delta_{\kappa\mu}(-\tau))$ connecting to time zero by placing $F_\mu^\dagger \Delta_{\mu\kappa}$ $(F_\kappa \Delta_{\kappa\mu})$ on its left vertex, and $F_\kappa$ $(F_\mu^\dagger)$ at the zero vertex. For (27), this is the first equality in (31).

4. Select one of the remaining hybridization lines $\Delta_{\pi\nu}$ $(\Delta_{\nu\pi})$, and split the current diagram into two sums, containing $p$ terms in total: one corresponding to non-positive SOE expansion frequencies $\omega_l \leq 0$, and the other to positive frequencies $\omega_l > 0$.

    (a) For term $l$ of the $\omega_l \leq 0$ sum, place $K_l^+ \overline{F}_{\nu l}^\dagger$ $(K_l^- \overline{F}_{\pi l})$ at the left vertex of the hybridization line, $K_l^- F_\nu$ $(K_l^+ F_\pi^\dagger)$ at the right vertex, and divide by $K_l^-(0)$ $(-K_l^+(0))$.

    (b) For term $l$ of the $\omega_l > 0$ sum, place $\overline{F}_{\nu l}^\dagger$ $(\overline{F}_{\pi l})$ at the left vertex of the hybridization line, $F_\nu$ $(F_\pi^\dagger)$ at the right vertex, $K_l^+$ $(K_l^-)$ on each backbone edge between the two vertices, and divide by $(K_l^+(0))^j$ $((-K_l^-(0))^j)$, where the number of edges between the left and right vertices is $j + 1$.

    For (27), this is the second equality in (31).

5. Repeat Step 4 for each remaining hybridization line in each diagram, introducing orbital and SOE expansion indices for each one. For (27), this is the third equality in (31), repeated once to separate $\Delta_{\xi\lambda}$.

This systematic procedure can be used to decompose diagrams of any order, and generates sums over $2^{m-1}$ distinct types of backbone diagrams (four in (31)). Each of these backbone diagrams can be constructed by accumulating the functions associated with each vertex and edge of each backbone diagram via multiplication on the DLR imaginary time grid. The sums over the diagram types and the pole indices $\omega_l, \omega_{l'}, \dots$ are performed explicitly.

### C. Evaluation of backbone diagrams and computational complexity

We must next evaluate each of the backbone diagrams produced by the decomposition procedure above, and sum the results. We note that the sums over all $p^{m-1}$ combinations of SOE expansion indices ($l$ and $l'$ in (31)) and $n^{m-1}$ combinations of impurity state indices not associated with the un-decomposed hybridization line ($\lambda$ and $\nu$ in (31)) are performed explicitly and in parallel. We can therefore consider all indices fixed, except for the impurity state indices associated with the un-decomposed hybridization line ($\mu$ and $\kappa$ in (31)).

The backbone diagram evaluations then proceed in the same manner as for the OCA self-energy example above:

1. Starting from $\tau_1$, proceed right to left, performing multiplications at vertices and convolutions at edges, until reaching the vertex containing the un-decomposed hybridization line $\Delta_{\mu\kappa}$.

2. For each $\kappa$, multiply by $F_\kappa$ $(F_\kappa^\dagger)$. Then, for each $\mu, \kappa$, multiply by $\Delta_{\mu\kappa}$, and sum over $\kappa$. Finally, for each $\mu$, multiply by $F_\mu^\dagger$ $(F_\mu)$ and sum over $\mu$.

3. Continue right to left until the final vertex multiplication is complete.

Steps 1 and 3 involve multiple types of operations, all of which appear in the OCA self-energy example in Sec. IV A, where their computational complexities are stated. For completeness, we list all such operations, and their corresponding complexities: (i) multiplication of an $N \times N$ matrix-valued function by an $N \times N$ matrix or another $N \times N$ matrix-valued function ($\mathcal{O}(rN^3)$), (ii) multiplication of a scalar-valued function by another scalar-valued function ($\mathcal{O}(r)$) or an $N \times N$ matrix-valued function ($\mathcal{O}(rN^2)$), and (iii) convolution of two $N \times N$ matrix-valued functions ($\mathcal{O}(r^2N^3)$). The cost of the convolution operation dominates, and $\mathcal{O}(m)$ convolutions appear in each backbone diagram. Step 2 is identical to Steps 3-5 in the example from Sec. IV A, and has the same computational complexity ($\mathcal{O}(nrN^3)$). We therefore arrive at the total computational complexity

$$\mathcal{O}((np)^{m-1}(mr^2N^3 + nrN^3)) \tag{32}$$

for evaluation of all $C(m)$ decomposed self-energy diagrams at order $m$ for a given topology [61, 62] and $2^m$ combinations of forward and backward-propagating hybridization lines. It includes the negligible cost of pre-computing the quantities $\overline{F}_{kl}^\dagger$ and $\overline{F}_{kl}$. We see that (32) correctly reduces to (24) when $m = 2$.

This can be compared with the $\mathcal{O}(mn^{2m}r^{m+1}N^3)$ complexity of the algorithm presented in Ref. 43. We always have $p \leq r$, and these two parameters were compared for several examples in Sec. III D. The reduction is achieved by (i) the use of the procedure described in Sec. III rather than the DLR for hybridization fitting, and (ii) a rearrangement of the order of summation over impurity state indices. An analogous decomposition and evaluation procedure for single-particle Green's function diagrams is described in App. B. There are only minor modifications compared with the pseudo-particle self-energy diagrams, and the computational complexity is the same.

## V. NUMERICAL RESULTS

We demonstrate the performance of our algorithm on several examples: a simple spinless fermion dimer coupled to a discrete bath, a two-band impurity model with both discrete and metallic baths, and a self-consistent

DMFT study of strongly correlated magnetism for a minimal three-orbital model of $Ca_2RuO_4$ including spin-orbit coupling. Our numerical results are generated using a code based on the `cppdlr` [83] and `triqs` [84] libraries. We perform hybridization fitting using the `adapol` library [85, 86], and build the local Hilbert space for the $Ca_2RuO_4$ model using the `pyed` library [87]. We use MPI to parallelize the diagrammatic sums over impurity state, hybridization propagation direction, and SOE expansion indices, yielding near-perfect parallel efficiency.

Solving the pseudo-particle Dyson equation to obtain $\mathcal{G}$ can be numerically unstable at low temperatures, since it supports exponentially growing solutions. To address this, we use a self-consistent iteration, adjusting the pseudo-particle chemical potential $\eta$ at each step using Newton's method with explicitly-computed derivative information. The details of our approach are described in Apps. C and E.

## A. Fermionic dimer

We validate the accuracy and order-by-order convergence of our solver for a fermionic spinless dimer coupled to a discrete bath with off-diagonal hybridization:

$$\hat{H} = \hat{H}_{\text{dimer}} + \hat{H}_{\text{bath}} + \hat{H}_{\text{coupling}},$$
$$\hat{H}_{\text{dimer}} = -v(\hat{c}_0^\dagger \hat{c}_1 + \hat{c}_1^\dagger \hat{c}_0) + U\hat{n}_0\hat{n}_1,$$
$$\hat{H}_{\text{bath}} = -t_{\text{b}} \sum_{k=0}^{1}(\hat{b}_{0k}^\dagger \hat{b}_{1k} + \hat{b}_{1k}^\dagger \hat{b}_{0k}), \quad (33)$$
$$\hat{H}_{\text{coupling}} = -t \sum_{\lambda=0}^{1} \sum_{k=0}^{1}(\hat{c}_\lambda^\dagger \hat{b}_{\lambda k} + \hat{b}_{\lambda k}^\dagger \hat{c}_\lambda).$$

Here $\hat{c}_\lambda$ is the fermionic annihilation operator for the impurity state $\lambda$, $\hat{n}_\lambda = \hat{c}_\lambda^\dagger \hat{c}_\lambda$ is the corresponding number operator, and $\hat{b}_{\lambda k}$ is the fermionic annihilation operator for the $k$th bath state coupled to the impurity state $\lambda$. The parameters $v$, $t_{\text{b}}$, and $t$ are the hopping strengths for the impurity, the bath, and the impurity-bath coupling, respectively, and $U$ is the interaction strength between the impurity states. Following Refs. 43 and 48, we take $t = 1$, $t_{\text{b}} = 1.5t$, $v = 1.5t$, and $U = 4t$.

We compute the single-particle Green's function $G$ by solving the pseudo-particle Dyson equation self-consistently, using a strong coupling expansion of various orders. In this case, the hybridization function defined by (33) is given explicitly by $\Delta(i\nu_n) = 2t^2(i\nu_n I - H_0)^{-1}$, where $H_0 = \begin{pmatrix} 0 & t_{\text{b}} \\ t_{\text{b}} & 0 \end{pmatrix}$ and $I$ is the $2 \times 2$ identity matrix. Our hybridization fitting algorithm identifies the $p = 2$ poles of $\Delta$ (eigenvalues of $H_0$), and since our diagram evaluation algorithm scales as $\mathcal{O}((np)^{m-1})$, we are able to handle relatively high expansion orders $m$ even for very low temperatures. This is therefore a significantly easier case for our algorithm than the continuous spectra discussed later.

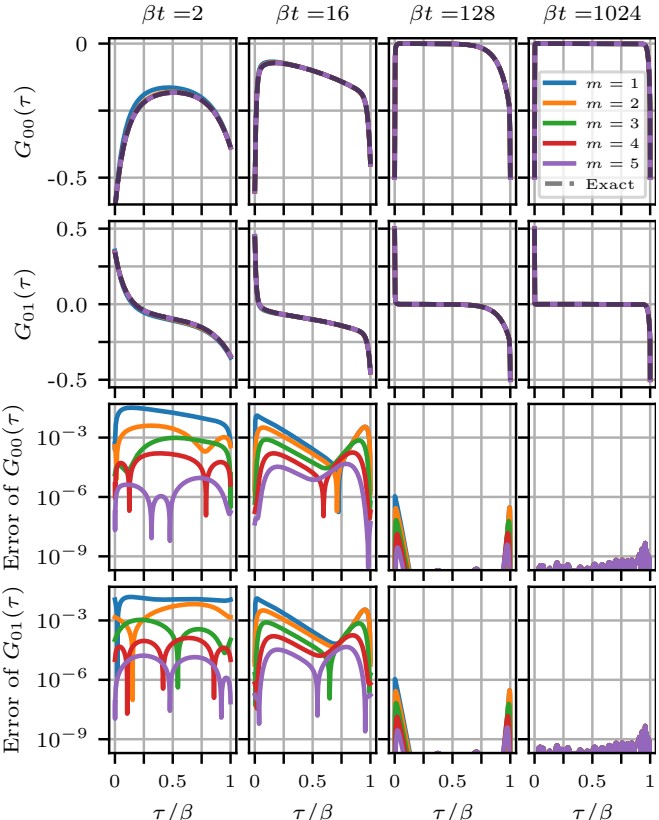

FIG. 3. Single-particle Green's function for the fermionic spinless dimer model (33), at several temperatures and strong coupling expansion orders, along with pointwise errors.

In Fig. 3, we compare $G_{00}$ and $G_{01}$ to a solution obtained by exact diagonalization, at expansion orders $m = 1, \ldots, 5$, for inverse temperatures $\beta = 2, 16, 128, 1024$. We use the DLR parameters $\Lambda = 20\beta$, $\epsilon = 10^{-10}$. We use the self-consistency tolerance $10^{-9}$ on the pseudo-particle propagator. We observe a uniform improvement in accuracy as the expansion order is increased. The reduction in error with increasing $\beta$ is a consequence of the discrete spectrum of the bath. The hybridization function has a spectral gap at the Fermi level and at large $\beta$ it decays exponentially over a time interval proportional to the gap. This exponentially suppresses the contributions of concurrent hybridization events. Since the first-order ($m = 1$) approximation contains all possible sequential hybridization events, it is sufficient to describe the effect of the spectrally gapped bath on the dimer at low temperatures.

In Fig. 4(a), we plot the wall-clock timings for a single evaluation of all diagrams at a given order $m$ (including all topologies, hybridization propagation directions, and SOE expansion terms), for both the single-particle Green's function $G$ and the self-energy $\Sigma$, at $\beta = 16$. We observe close agreement with the expected computational complexity (32). In Fig. 4(b), we plot the $L^2$ error (17) of the single-particle Green's function with increasing expansion order $m$ at several $\beta$, and observe

exponential convergence (the flat error for $\beta t = 1024$ is a consequence of the pseudo-particle self-consistency tolerance). At third-order and beyond, we achieve errors smaller than the stochastic noise reported in the inchworm Monte Carlo study in Ref. 48 for the same model, which had a wall-clock timing of 500 core-hours for $\beta = 16$. By contrast, our third, fourth, fifth, and sixth-order calculations for $\beta = 16$ took 3 core-seconds, 68 core-seconds, 1.2 core-hours, and 95 core-hours, respectively, which could be further improved by relaxing the DLR and pseudo-particle self-consistency tolerances.

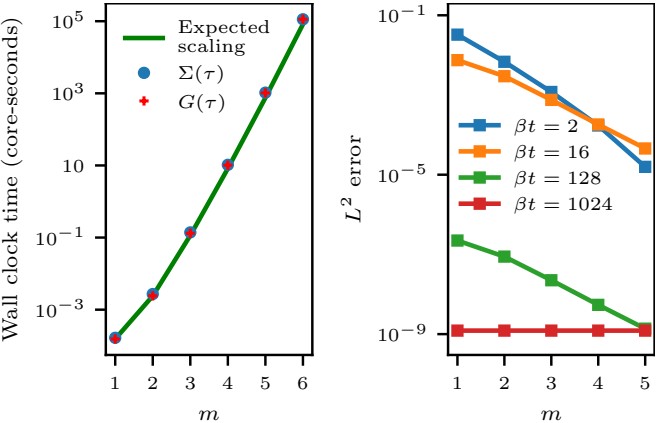

FIG. 4. (a) Wall clock timings for a single evaluation of all pseudo-particle self-energy ($\Sigma$) and single-particle Green's function ($G$) diagrams at a given order, for inverse temperature $\beta = 16$. The expected scaling is $\mathcal{O}(C(m)2^{m-1}(np)^{m-1}(mr^2N^3 + nrN^3))$, where $C(m)$ is the number of diagram topologies at expansion order $m$, and we choose the prefactor to match the $m = 1$ data point. (b) $L^2$ error (17) of $G$ versus $m$ at various temperatures.

## B. Two-band $e_g$ model

The correlated $e_g$ bands in transition metal oxides are a prototypical example of a multi-orbital system with strong correlations. We consider the following two-band Anderson impurity model, in which the local electronic Coulomb interaction takes the Kanamori form [88]:

$$\hat{H}_{\text{loc}} = U \sum_{\kappa=0}^{1} \hat{n}_{\kappa\uparrow}\hat{n}_{\kappa\downarrow} + \sum_{\sigma,\sigma'\in\{\uparrow,\downarrow\}} (U' - J_H\delta_{\sigma\sigma'})\hat{n}_{0\sigma}\hat{n}_{1\sigma'}$$
$$+ J_H \sum_{\kappa\neq\lambda\in\{0,1\}} \left( \hat{c}_{\kappa\uparrow}^\dagger\hat{c}_{\kappa\downarrow}^\dagger\hat{c}_{\lambda\downarrow}\hat{c}_{\lambda\uparrow} + \hat{c}_{\kappa\uparrow}^\dagger\hat{c}_{\lambda\downarrow}^\dagger\hat{c}_{\kappa\downarrow}\hat{c}_{\lambda\uparrow} \right).$$
$$(34)$$

Here $\hat{c}_{\kappa\sigma}$ is the fermionic annihilation operator for the $\kappa$th orbital with spin $\sigma$ and $\hat{n}_{\kappa\sigma} = \hat{c}_{\kappa\sigma}^\dagger\hat{c}_{\kappa\sigma}$ is the corresponding number operator. $U$ is the Hubbard interaction strength, $J_H$ is the Hund's coupling strength, and $U' = U - 2J_H$. Following Refs. 43 and 48 we consider the system-bath coupling given by the hybridization (14) with $\rho_{\kappa\lambda}(\omega) = (\delta_{\kappa\lambda} + s(1 - \delta_{\kappa\lambda})) t^2 J(\omega)$, where $s$ is the strength of the

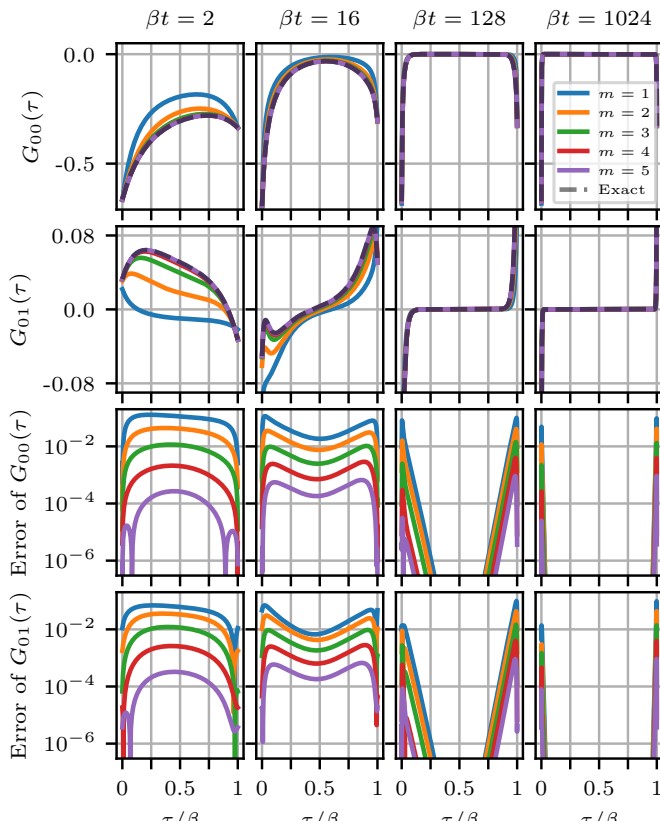

FIG. 5. Single particle Green's function for the two-band $e_g$ model with a discrete bath at several temperatures and strong coupling expansion orders, along with pointwise errors.

off-diagonal hybridization, and $J(\omega)$ is to be specified. The chemical potential is set to be $\mu = (3U - 5J_H)/2 + \Delta\mu$, where $\Delta\mu = -1.5$.

We first consider a discrete bath with a single bath site per impurity orbital: $J(\omega) = \sum_{k=0}^{1} \delta(\omega - \epsilon_k)$, with $\epsilon_0 = -2.3t$, $\epsilon_1 = 2.3t$. We take $U = 2.0$, $J_H = 0.2$, and $s = 0.5$, which yields a strong off-diagonal hybridization. We use the DLR parameters $\Lambda = 10\beta$ and $\epsilon = 10^{-10}$, and a pseudo-particle self-consistency tolerance $10^{-6}$. Fig. 5 shows the self-consistently computed single-particle Green's function $G_{\kappa\lambda}(\tau)$ at expansion orders $m = 1, \ldots, 5$ and inverse temperatures $\beta = 2, 16, 128, 1024$, as well as the error compared to a reference computed by exact diagonalization. We observe regular order-by-order convergence at each temperature.

We next consider a metallic bath, $J(\omega) = \frac{2}{\pi D^2}\sqrt{D^2 - \omega^2}$, with $D = 2t$, $s = 1$, and $\beta = 8$. We use the DLR parameters $\Lambda = 10\beta$ and $\epsilon = 10^{-6}$, and a pseudo-particle self-consistency tolerance $10^{-4}$. The metallic bath has a continuum of states at zero energy, enabling large quantum fluctuations between the local atomic states and the bath. The zero energy cost of bath fluctuations drives a large number of concurrent hybridization events. This case is therefore challenging for methods based on a hybridization expansion, like ours.

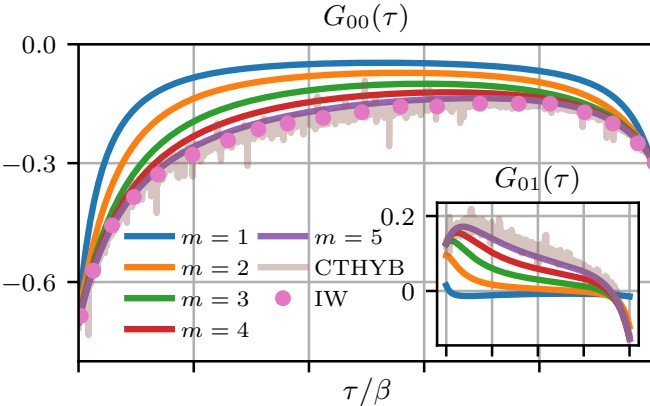

FIG. 6. Single-particle Green's function for the two-band $e_g$ model with a metallic bath at inverse temperature $\beta t = 8$, obtained using our method at various orders, as well as inchworm Monte Carlo results reported in Ref. 48, and CT-HYB results reported in Ref. 43.

In Ref. 43, we showed that a third-order strong coupling expansion was not well-converged with respect to the expansion order for this system, and that the CT-HYB method has a significant sign problem. With our improved algorithm, we are able to reach fifth-order at a modest cost of 750 core-hours (compared to 150 core-hours for the third-order result using the previous algorithm; using our algorithm, the third-order calculation takes approximately one core-minute). Fig. 6 compares our order-by-order results with those of CT-HYB [9] and inchworm Monte Carlo, reported in Ref. 48 (note that inchworm results are not reported for the off-diagonal components). We observe that a fifth-order expansion provides a significant improvement over a third-order expansion, yielding a solution nearly in agreement with the inchworm Monte Carlo result, but free from Monte Carlo noise. The wall clock timing for the inchworm Monte Carlo result reported in Ref. 48 is 1500 core-hours.

### C. Strongly correlated magnetism in $Ca_2RuO_4$

The minimal model of $Ca_2RuO_4$ used as a benchmark in Ref. 43 is a realistic example of a system amenable to strong coupling calculations using our method. In that reference, we performed a self-consistent DMFT calculation of the single-particle Green's function in the paramagnetic phase at the high temperature $T = 1160\,\mathrm{K}$ ($\beta = 10\,\mathrm{eV}^{-1}$), up to third-order in the strong coupling expansion. The calculation used the DLR parameters $\Lambda = 100$ and $\epsilon = 10^{-6}$, yielding 26 basis functions (in that work, used both for hybridization fitting and imaginary time operations), and the pseudoparticle self-energy and DMFT lattice self-consistencies were performed in tandem, with a tolerance threshold of $10^{-6}$ for changes in the respective propagators between iterations. The first-, second- and third-order calculations required 11,

9 and 7 iterations, respectively, using the solution from the previous order as an initial guess. The total calculation required 90,000 core-hours on a large compute cluster. Using the improved algorithm presented in this paper, this calculation requires less than 40 core-hours (24 minutes on a single 96-core node) using the same DLR parameters and self-consistent iteration tolerances. This shows the effectiveness of our bold strong coupling expansion algorithm particularly for strongly correlated systems in the Mott insulating and magnetically ordered states with low symmetry. The modest computational cost allows us to study the temperature dependence of the model, in particular the anti-ferromagnetic transition, as well as the effect of spin-orbit coupling and the expansion order.

We first briefly describe our minimal model of $Ca_2RuO_4$ [43, 89], an effective three-band low energy model filled with four electrons. The bands originate from three local orbitals with Ru $4d$-$t_{2g}$ cubic harmonic symmetry, denoted as $xy$ for the in-plane orbital and $xz$, $yz$ for the two out-of-plane orbitals. The lattice dispersion—which, for a quantitative study, should be constructed from an ab-initio calculation—is modeled using a Bethe lattice semi-circular density of states with nearest neighbour hopping $t_{xy} = 0.5\,\mathrm{eV}$ in-plane and $t_{xz} = t_{yz} = 0.25\,\mathrm{eV}$ out-of-plane, to roughly match the ab-initio bandwidths [90]. The effect of the layered tetragonal structure is accounted for by the crystal field $\Delta_{\mathrm{cf}} = 0.5\,\mathrm{eV}$, lowering the $xy$-orbital onsite energy. We note that the time-dependent dynamics of the model were also studied in Ref. 89, but without spin-orbit coupling and using only a first-order expansion.

The structure of the local Coulomb interaction is obtained by projection on the orbitals with Ru $4d$-$t_{2g}$ cubic harmonic symmetry, producing the Kanamori interaction [91, 92]

$$H_{\mathrm{int}} = U \sum_a \hat{n}_{a\uparrow}\hat{n}_{a\downarrow} + \frac{1}{2}\sum_{a\neq b}\sum_{\sigma,\sigma'}\left(U' - J\delta_{\sigma\sigma'}\right)\hat{n}_{a\sigma}\hat{n}_{b\sigma'}$$
$$- \sum_{a\neq b}\left(J\hat{c}^\dagger_{a\uparrow}\hat{c}_{a\downarrow}\hat{c}^\dagger_{b\downarrow}\hat{c}_{b\uparrow} + J'\hat{c}^\dagger_{b\uparrow}\hat{c}^\dagger_{b\downarrow}\hat{c}_{a\uparrow}\hat{c}_{a\downarrow}\right), \quad (35)$$

where the density operator is given by $\hat{n}_{a\sigma} = \hat{c}^\dagger_{a\sigma}\hat{c}_{a\sigma}$, with spin ($\sigma \in \{\uparrow,\downarrow\}$) and orbital ($a,b \in \{xz,yz,xy\}$) indices. We adopt the established values $U = 2.3\,\mathrm{eV}$, $J = 0.4\,\mathrm{eV}$ [90], and the rotationally invariant form of the Kanamori interaction with $U' = U - 2J$ and $J' = J$.

The spin-orbit interaction projected on the $4d - t2g$ subset of cubic harmonics is given by [93]

$$H_{\mathrm{soc}} = \lambda_{\mathrm{soc}} \sum_{ij} \hat{\Psi}^\dagger_i [h_{\mathrm{soc}}]_{ij}\hat{\Psi}_j, \quad (36)$$

where $\hat{\Psi} = [\hat{c}_{xz\uparrow}, \hat{c}_{yz,\uparrow}, \hat{c}_{xy,\downarrow}, \hat{c}_{xz\downarrow}, \hat{c}_{yz,\downarrow}, \hat{c}_{xy,\uparrow}]$ and $h_{\mathrm{soc}}$ is

the matrix

$$h_{soc} = \begin{pmatrix} \begin{array}{ccc|ccc} 0 & -i & i & & & \\ i & 0 & -1 & & 0 & \\ -i & -1 & 0 & & & \\ \hline & & & 0 & i & i \\ & 0 & & -i & 0 & 1 \\ & & & -i & 1 & 0 \end{array} \end{pmatrix}. \tag{37}$$

Following Ref. [43] we adopt the spin-orbit coupling strength $\lambda_{soc} = 0.1\,\mathrm{eV}$, consistent with estimates from resonant inelastic X-ray scattering (reporting $\lambda_{soc} \approx 0.13\,\mathrm{eV}$) [94].

In order to capture the symmetry breaking on the bipartite Bethe lattice, we solve two impurity problems, one for each class of lattice sites (A and B), coupled through the DMFT self-consistency relation

$$\begin{aligned} \Delta_A(\tau) &= \mathbf{t} \cdot G_B(\tau) \cdot \mathbf{t}, \\ \Delta_B(\tau) &= \mathbf{t} \cdot G_A(\tau) \cdot \mathbf{t}. \end{aligned} \tag{38}$$

Here the hybridization functions $\Delta_{\{A,B\}}(\tau)$, the single-particle Green's functions $G_{\{A,B\}}(\tau)$, and the hopping $\mathbf{t} = \mathrm{diag}(t_{xz}, t_{yz}, t_{xy}, t_{xz}, t_{yz}, t_{xy})$ are matrices in spin-orbital space. We note that the spin-orbit coupling breaks $SU(2)$ spin-symmetry, and complicates a simple reformulation of the self-consistency in terms of a single impurity model. Using two separate impurities enables us to explore magnetic symmetry breaking in arbitrary directions, and in particular spin-orbit-induced anisotropy between the in-plane and out-of-plane axes.

We solve the DMFT self-consistency (38) and pseudo-particle self-consistency in tandem using the method described in Apps. C and E. Thus, given a guess of the hybridization function $\Delta$ and the pseudo-particle self-energy $\Sigma$, we compute the pseudo-particle Green's function $\mathcal{G}$ via the Dyson equation and the single-particle Green's function $G$ via its diagrammatic expansion, and then use them to update $\Sigma$ via its diagrammatic expansion and $\Delta$ via (38). We use a self-consistency tolerance $10^{-6}$ on $\mathcal{G}$ and $\Delta$. In the high temperature paramagnetic phase, we extend this idea to fixed density calculations, also adjusting the physical chemical potential $\mu$ using explicitly-computed derivatives, as described in Apps. D and E.

At elevated temperatures the system is a paramagnetic Mott insulator, with half-filled out-of-plane orbitals $xz$, $yz$ ($\langle\hat{n}_{xz,\sigma}\rangle = \langle\hat{n}_{yz,\sigma}\rangle \approx 0.5$) having a distinct Mott gap, and the in-plane orbital $xy$ completely filled ($\langle\hat{n}_{xy,\sigma}\rangle \approx 1$). As the temperature is lowered the system undergoes a second-order phase transition to an anti-ferromagnet. The bipartite symmetry breaking in spin originates mainly from the out-of-plane half-filled orbitals, which remain (approximately) half-filled while becoming spin-polarized at the phase transition. These observations are summarized in Fig. 7.

One self-consistent DMFT and pseudo-particle iteration for a third-order expansion takes approximately 2.2 core-hours, and approximately 10 iterations are required

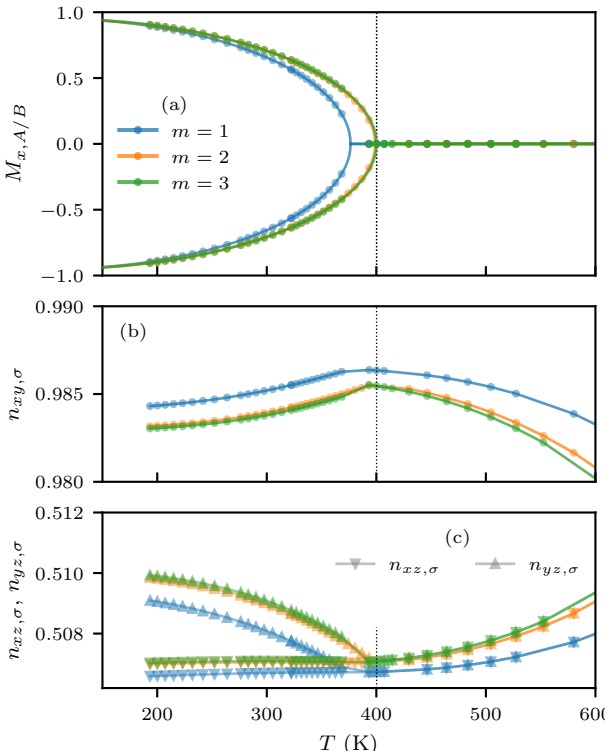

FIG. 7. Anti-ferromagnetic transition in the $Ca_2RuO_4$ three-band model with spin-orbit coupling. (a) Magnetization $M_x = \langle\hat{S}_x\rangle$ on the $A$ (positive values) and $B$ (negative values) sublattices, as a function of temperature $T$ for various strong coupling expansion orders. The solid lines are mean-field critical exponent fits, and from the third-order expansion results we extrapolate the transition temperature $T_N \approx 400\,\mathrm{K}$ (vertical dotted line). (b) Filling $n_{xy,\sigma}$ of the in-plane $xy$ orbital (per spin $\sigma \in \{\uparrow,\downarrow\}$). (c) Filling of the out-of-plane orbital parallel ($xz$, down triangle) and perpendicular ($yz$, up triangle) to the magnetic symmetry breaking, here along the $x$-axis ($M_{y,A/B} = M_{z,A/B} = 0$).

far from the magnetic phase transition. However, the fixed point iteration exhibits a critical slowing down approaching the second-order phase transition, associated with the bifurcation of solutions [92, 95]. Quasi-Newton methods can in principle be used to overcome this, as has been done for the single band Hubbard model [95]. Below the transition temperature $T_N$ the paramagnetic state can still be found as a metastable solution, as we observe at third order (see the green markers with $M_{x,A/B} = 0$ below $T_N$ in Fig. 7).

As reported previously [43], in the high temperature paramagnetic phase, the single-particle Green's function converges rapidly with the strong coupling expansion order. Fig. 7 shows that this remains true for static expectation values down to and through the anti-ferromagnetic phase transition. While the qualitative picture of the paramagnetic phase holds when neglecting spin-orbit coupling, we are now able to characterize the fine details in the mixing of spin and orbital degrees of freedom due

to non-zero spin-orbit coupling. The in-plane $xy$ orbital is lightly hole-doped ($\langle \hat{n}_{xy,\sigma} \rangle \approx 0.9850$) and the out-of-plane orbitals are correspondingly lightly electron-doped ($\langle \hat{n}_{xz,\sigma} \rangle = \langle \hat{n}_{yz,\sigma} \rangle \approx 0.5075$). The doping is further enhanced at higher temperatures.

The spin-orbit coupling also has a significant effect on the magnetic ordering in the anti-ferromagnetic state. The coupling between spin and orbital degrees of freedom destroys the magnetic $SU(2)$ symmetry and the system preferentially orders anti-ferromagnetically in the $xy$-plane. In the calculation presented in Fig. 7 the order has been seeded by a small applied field in the $x$-direction in the initial guess of the self-consistent iteration. We find that seeding the calculation in the $z$-direction is unstable and eventually collapses to an in-plane $xy$ order (in a random direction). The magnetic symmetry breaking lifts, in turn, the degeneracy of the two out-of-plane orbitals ($xz$ and $yz$), as can be seen in the lower panel in Fig. 7. The presence of the in-plane $xy$ order aligns with the analysis in Ref. 96, which utilized an effective low-energy magnetic mode to attribute the in-plane order to a significant asymmetry in the single-ion anisotropy tensor, primarily dominated by the in-plane components. This in-plane order is also consistent with magnetic susceptibility measurements [97] and neutron scattering experiments [98]. However, determining the precise direction of the in-plane magnetization is a subtler issue, which would require ab-initio modeling and consideration of structural distortions——an important problem for future research.

To determine the Néel temperature $T_N$ of the transition, we performed a least squares fit of the computed magnetic orders (markers in the upper panel of Fig. 7) to a mean-field asymptotic square root $p(T-T_N)\sqrt{T-T_N}$, for $p$ a cubic polynomial, and minimized the residual with respect to $T_N$. This yields an estimate $T_N \approx 400$ K. The experimental Néel temperature of $Ca_2RuO_4$ is approximately 110 K [97, 98]. The discrepancy can be attributed to the freezing of spatial fluctuations within DMFT and the lack of quantitative treatment of the single-particle dispersion.

## VI. CONCLUSION

We have presented a deterministic evaluation algorithm for bold pseudo-particle strong coupling diagrams, along with realistic examples of its use as a low-cost quantum impurity solver. Using efficient hybridization fitting and precomputation of certain quantities, we have achieved orders of magnitude performance improvements over the similar algorithm of Ref. 43. We have described an automated parallel implementation of the full procedure for multi-orbital systems using arbitrary-order expansions. We plan to make this implementation available as an open source software package in the near future.

This approach enables the exploration of novel physics in materials for which strong spin-orbit coupling plays a crucial role, such as in relativistic Mott insula-

tors. Notable examples include $Ca_2RuO_4$ [90, 96, 99], $Sr_2IrO_4$ [100–102], and $Nb_3Cl_8$ [103], in which the interplay between electronic correlations, spin-orbit coupling, and lattice distortions gives rise to unconventional magnetic phases. A significant next step would be to provide our quantum impurity solver with ab-initio input using DFT+DMFT [104], allowing for a more accurate and comprehensive description of these materials. An important advantage of our approach is that it can be used to study the effect of strong correlations at very low temperatures, which could provide valuable insights into the nature of low temperature ordered phases such as exotic magnetic phases.

Further algorithmic performance improvements remain to be explored. For example, incorporating Hamiltonian symmetries would replace the dense matrix-matrix multiplications in the local many-body space with sparse operations, leading to a significant cost reduction for systems with many orbitals. Extension of our method to real-time diagrammatics is also possible [60], in particular for strong coupling expansions of impurity models in equilibrium or steady state. After replacing the imaginary frequency hybridization fitting step with a suitable real frequency analogue, of which several have been proposed [60, 78–80, 105], the rest of our approach is straightforwardly generalized, using suitable algorithms for efficient operations on one-dimensional real time grids. These directions are topics of our current research.

## ACKNOWLEDGMENTS

This work is partially supported by the Simons Targeted Grant in Mathematics and Physical Sciences on Moiré Materials Magic (ZH). HURS acknowledges financial support from the Swedish Research Council (Vetenskapsrådet, VR) grant number 2024-04652 and funding from the European Research Council (ERC) under the European Union's Horizon 2020 research and innovation programme (grant agreement No. 854843-FASTCORR). Some computations were enabled by resources provided by the National Academic Infrastructure for Supercomputing in Sweden (NAISS) through the projects NAISS 2024/8-15, NAISS 2024/1-18, and NAISS 2024/6-127 at PDC, NSC, and CSC, partially funded by the Swedish Research Council through grant agreements No. 2022-06725 and No. 2018-05973. DG is supported by the Slovenian Research and Innovation Agency (ARIS) under Programs No. P1-0044, No. J1-2455, and No. MN-0016-106. The Flatiron Institute is a division of the Simons Foundation.

## Appendix A: Symmetrization of AAA hybridization fitting

This Appendix describes our method to enforce the symmetry relation $r(-i\nu_n) = r^\dagger(i\nu_n)$ of the AAA ra-

tional approximation of a matrix-valued hybridization $\Delta(i\nu_n)$. If $z_j = i\nu_{n_j}$ is selected as a support point, then we automatically also select $\overline{z_j} = -i\nu_{n_j}$ as the next support point. In other words, at the $k$th iteration, we select a pair of support points $\{i\nu_{n_j}, -i\nu_{n_j}\}$, and then $\mathcal{Z}^{(k)} = \mathcal{Z}^{(k-1)} \cup \{i\nu_{n_j}, -i\nu_{n_j}\}$ comprises the new collection of $2k$ support points. We further enforce that $w^{(k)} = (w_1, \overline{w_1}, \ldots, w_k, \overline{w_k})$, i.e., that the weights corresponding to $i\nu_{n_j}$ and $-i\nu_{n_j}$ are complex conjugates. It follows immediately from the barycentric formula (8) that these two conditions imply the symmetry relation.

The latter condition requires replacing our SVD procedure to solve (10), which does not necessarily produce weights precisely in conjugate pairs even for conjugate paired $z_j$. To do so, we define matrices $A^{\pm,\nu\lambda} \in \mathbb{C}^{k \times k}$ by

$$A_{ij}^{\pm,\nu\lambda} = \frac{\Delta_{\nu\lambda}(\zeta_i) - \Delta_{\nu\lambda}(\pm i\nu_{n_j})}{\zeta_i \mp i\nu_{n_j}}, \qquad \text{(A1)}$$

for impurity indices $\nu, \lambda = 1, \ldots, n$, $\zeta_i \in Z \backslash \mathcal{Z}^{(k)}$, and $j = 1, \ldots, k$. Then, in analogy to (11), the weights are determined by solving the minimization problem

$$\min_{w_1,\ldots,w_k \in \mathbb{C}} \sum_{\nu,\lambda=1}^{n} \sum_{i=1}^{k} \left| \sum_{j=1}^{k} A_{ij}^{+,\nu\lambda} w_j + \sum_{j=1}^{k} A_{ij}^{-,\nu\lambda} \overline{w_j} \right|^2$$

$$\text{subject to} \quad \sum_{j=1}^{k} |w_j|^2 = 1. \qquad \text{(A2)}$$

Decomposing into real and imaginary parts, this can be rewritten as

$$\min_{u,v \in \mathbb{R}^k} \sum_{\nu,\lambda=1}^{n} \left\| \mathcal{A}^{\nu\lambda} \begin{pmatrix} u \\ v \end{pmatrix} \right\|_2^2 = \left\| \mathcal{A} \begin{pmatrix} u \\ v \end{pmatrix} \right\|_2^2, \qquad \text{(A3)}$$

where $w_j = u_j + iv_j$,

$$\mathcal{A}^{\nu\lambda} = \begin{pmatrix} \mathrm{Re}(A^{+,\nu\lambda} + A^{-,\nu\lambda}) & -\mathrm{Im}(A^{+,\nu\lambda} - A^{-,\nu\lambda}) \\ \mathrm{Im}(A^{+,\nu\lambda} + A^{-,\nu\lambda}) & \mathrm{Re}(A^{+,\nu\lambda} - A^{-,\nu\lambda}) \end{pmatrix}, \qquad \text{(A4)}$$

and $\mathcal{A} \in \mathbb{R}^{2kn^2} \times 2k$ is obtained by vertically stacking the matrices $\mathcal{A}^{\nu\lambda}$ for $\nu, \lambda = 1, \ldots, n$. The solution of (A3) is obtained from the right singular vector of $\mathcal{A}$ corresponding to its smallest singular value, which is real since $\mathcal{A}$ is real. The vector $w^{(k)}$ can be straightforwardly recovered from $u$ and $v$ in the desired form.

## Appendix B: Single-particle Green's function diagrams

We first describe the basic structure of the single-particle Green's function diagrams, highlighting their differences with the pseudo-particle self-energy diagrams. We again refer to Ref. 43 for further details. Then we describe our algorithm for decomposing and evaluating general Green's function diagrams.

Green's function diagrams of order $m$ contain $m-1$ hybridization insertions, and at order $m$ there are again $C(m)$ diagram topologies [61, 62], and $2^{m-1}$ combinations of propagation directions of the hybridization interactions. We use the notation $G_{j,k}^{(m)}(\tau)$ to refer to the Green's function diagram of order $m$ with the $j$th topology and $k$th combination of hybridization directions, so that the single-particle Green's function is given by $G = \sum_{m=1}^{\infty} \sum_{j=1}^{C(m)} \sum_{k=1}^{2^{m-1}} G_{j,k}^{(m)}$. We note that the Green's function, as well as each of these diagrams, are $n \times n$ matrices. As an example, we consider the following third-order diagram:

$$G_{1,2,\nu\kappa}^{(3)}(\tau) = \tau, \nu \ \text{[diagram]} \ 0, \kappa = d_{312} \int_{\tau}^{\beta} d\tau_4 \int_{\tau}^{\tau_4} d\tau_3 \int_{0}^{\tau} d\tau_2 \int_{0}^{\tau_2} d\tau_1 \, \Delta_{\lambda\pi}(\tau_1 - \tau_4) \Delta_{\xi\mu}(\tau_3 - \tau_2)$$

$$\times \mathrm{Tr}\left[ \mathcal{G}(\beta - \tau_4) \, F_\lambda \, \mathcal{G}(\tau_4 - \tau_3) \, F_\xi^\dagger \, \mathcal{G}(\tau_3 - \tau) \, F_\nu \, \mathcal{G}(\tau - \tau_2) \, F_\mu \, \mathcal{G}(\tau_2 - \tau_1) \, F_\pi^\dagger \, \mathcal{G}(\tau_1 - 0) \, F_\kappa^\dagger \right]. \qquad \text{(B1)}$$

The prefactor $d_{mjk}$ is again $\pm 1$, analogous to $c_{mjk}$ for the pseudo-particle self-energy diagrams. These diagrams can be read in the same manner as the pseudo-particle self-energy diagrams, with a few key differences: (i) the diagrams are circular, with the final imaginary time $\beta$ identified with the initial time 0, (ii) the external time $\tau$ is inserted in the middle of the diagram, rather than the end, (iii) the vertices at time 0 and $\tau$ are not connected to a hybridization insertion, but they do have creation/annihilation matrix insertions, and (iv) a trace appearing in the integrand converts the result from an $N \times N$ matrix-valued function to a scalar-valued func-

tion. For the purposes of our decomposition procedure, it is convenient to "unroll" the circular diagram into a form closer to the pseudo-particle self-energy diagrams:

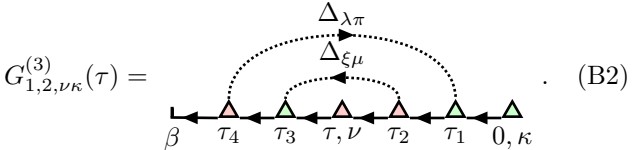

$$G^{(3)}_{1,2,\nu\kappa}(\tau) = \qquad\qquad . \quad (B2)$$

For the Green's function diagrams, all hybridization lines break the convolutional structure of the backbone and must be separated. We can directly apply the decomposition procedure described in Sec. IV B, with two modifications: (i) at the end of Step 2, we place $F^\dagger_\kappa$ on the $0, \kappa$ vertex, and $F_\nu$ on the $\nu$ vertex, corresponding to time $\tau$, and (ii) we skip Step 3, since it is not applicable. For the diagram (B2), the result is as follows:

$$d_{312}G^{(3)}_{1,2,\nu\kappa}(\tau) = \qquad$$

$$= -\sum_{\omega_l \le 0}\frac{1}{K^+_l(0)} \qquad$$

$$-\sum_{\omega_l > 0}\frac{1}{(K^-_l(0))^3} \qquad$$

$$= -\sum_{\omega_l \le 0, \omega_{l'} \le 0}\frac{1}{K^+_l(0)K^-_{l'}(0)} \qquad$$

$$-\sum_{\omega_l \le 0, \omega_{l'} > 0}\frac{1}{K^+_l(0)K^+_{l'}(0)} \qquad$$

$$-\sum_{\omega_l > 0, \omega_{l'} \le 0}\frac{1}{(K^-_l(0))^3 K^-_{l'}(0)} \qquad$$

$$-\sum_{\omega_l > 0, \omega_{l'} > 0}\frac{1}{(K^-_l(0))^3 K^+_{l'}(0)} \qquad .$$

$$(B3)$$

The resulting backbone diagrams can be converted to integral expressions just as before. For example, the second backbone diagram in the result is given by

$$-\sum_{\omega_l \leq 0, \omega_{l'} > 0} \frac{1}{K_l^+(0)K_{l'}^+(0)} \operatorname{Tr} \int_\tau^\beta d\tau_4 \, \mathcal{G}(\beta - \tau_4) K_l^-(\tau_4) \, \overline{F}_{\pi l} \int_\tau^{\tau_4} d\tau_3 \, \mathcal{G}(\tau_4 - \tau_3) \, \overline{F}_{\mu l'}^\dagger \, (\mathcal{G}K_{l'}^+)(\tau_3 - \tau)$$

$$\times F_\nu \int_0^\tau d\tau_2 \, (\mathcal{G}K_{l'}^+)(\tau - \tau_2) F_\mu \int_0^{\tau_2} d\tau_1 \mathcal{G}(\tau_2 - \tau_1) F_\pi^\dagger \, (\mathcal{G}K_l^+)(\tau_1) F_\kappa^\dagger. \tag{B4}$$

We note that the decomposed Green's function diagrams separate into the product of two sequences of backbone products and convolutions, one representing the portion from time 0 to $\tau$, and the other from time $\tau$ to $\beta$. Evaluation of backbone diagrams can be carried out in a similar manner to that described in Sec. IV C for the pseudo-particle self-energy diagrams, but with some key differences. We again loop through the $(np)^{m-1}$ combinations of SOE expansion and impurity state indices, except for the impurity state indices corresponding to the creation and annihilation matrices at times 0 and $\tau$ ($\kappa$ and $\nu$ above). For each, we do the following:

1. Evaluate the first sequence of backbone products and convolutions from $\tau_1$ to $\tau$, proceeding right to left, not including the creation and annihilation matrices at the endpoints. The result is an $N \times N$ matrix-valued function of $\tau$.

2. Evaluate the second sequence of backbone products and convolutions from $\tau$ to $\beta$ by transforming convolutions of the form $\int_\tau^\beta d\tau' \, F(\tau - \tau') G(\tau')$ into the usual form $\int_0^\tau d\tau' \, F(\tau - \tau') G(\tau')$ using a change of variables, as described in Ref. 43, App. A. The result is another $N \times N$ matrix-valued function of $\tau$.

3. For each $\nu$, perform the multiplication with the annihilation matrix $F_\nu$ at time $\tau$, as well as the scalar prefactor in the sum over SOE expansion indices, but leaving out the creation matrix $F_\kappa^\dagger$ at time 0. Accumulate the resulting $n$ $N \times N$ matrix-valued functions of $\tau$ in running sums over the $(np)^{m-1}$ SOE expansion and impurity state indices.

Once these steps are complete, the $(np)^{m-1}$ SOE expansion and impurity state indices will have been summed over, leaving $n$ $N \times N$ matrix-valued functions of $\tau$ indexed by $\nu$. Now,

4. Loop over $\nu$ and the time 0 creation matrix index $\kappa$. For each pair, carry out the remaining multiplications, and compute the trace.

Following the counting in Sec. IV C, we find that the computational complexity of the first three steps is $\mathcal{O}((np)^{m-1}(mr^2N^3 + nrN^3))$, and that of the final step is $\mathcal{O}(n^2rN^3)$, which is subdominant for $m > 1$. We arrive at the same computational complexity as for the pseudo-particle self-energy diagrams.

## Appendix C: Solution of the pseudo-particle Dyson equation at fixed self-energy

The atomic pseudo-particle propagator satisfies the Dyson equation

$$(-\partial_\tau - \hat{H}_{\text{loc}} - \eta_0 I)\mathcal{G}_0(\tau) = 0, \quad \mathcal{G}_0(0) = -I, \tag{C1}$$

where $\hat{H}_{\text{loc}}$ is the $N \times N$ local (impurity) Hamiltonian, $\eta_0$ is the the pseudo-particle chemical potential, and $I$ is the $N \times N$ identity matrix. The solution is given by

$$\mathcal{G}_0(\tau) \equiv -e^{-\tau(\hat{H}_{\text{loc}} + \eta_0 I)}, \tag{C2}$$

with the corresponding partition function

$$Z_0 = -\operatorname{Tr}\left[\mathcal{G}_0(\beta)\right]. \tag{C3}$$

While (C1) admits exponentially growing solutions, physical observables are independent of $\eta_0$, so $\eta_0$ can always be chosen so that the solution is exponentially decaying. To ensure this we choose $\eta_0$ so that $Z_0 = 1$:

$$\eta_0 = \frac{1}{\beta} \log \operatorname{Tr}\left[e^{-\beta \hat{H}_{\text{loc}}}\right]. \tag{C4}$$

For the Anderson impurity model, the hybridization of the atomic degrees of freedom with the environment produces a pseudo-particle self-energy $\Sigma$ describing environmental fluctuations [43]. The corresponding Dyson equation for the pseudo-particle Green's function takes the form

$$(-\partial_\tau - \hat{H}_{\text{loc}} - (\eta_0 + \eta)I - \Sigma *)\mathcal{G} = 0, \quad \mathcal{G}(0) = -I, \tag{C5}$$

with the convolution operator defined as

$$(\Sigma * \mathcal{G})(\tau) = \int_0^\tau d\bar{\tau} \, \Sigma(\tau - \bar{\tau})\mathcal{G}(\tau). \tag{C6}$$

This integrodifferential equation of motion can be recast in integral form using (C1):

$$(I - \eta \, \mathcal{G}_0 * - \mathcal{G}_0 * \Sigma *)\mathcal{G} = \mathcal{G}_0. \tag{C7}$$

Solving (C7) with a general self-energy $\Sigma$ can produce exponentially growing solutions, and is vulnerable to numerical overflow. To remedy this we include the additional constraint $Z = -\operatorname{Tr}[\mathcal{G}(\beta)] = 1$ by varying $\eta$, determining it using Newton's method as a root finder. Imposing this constraint directly is prone to numerical

overflow, since the partition function $Z$ varies exponentially with $\eta$. To avoid this, we rewrite the constraint in terms of the free energy $\Omega$ as

$$\Omega(\eta) \equiv -\frac{1}{\beta} \log Z(\eta) = \frac{1}{\beta} \log \mathrm{Tr}\left[\mathcal{G}(\beta)\right] = 0, \qquad \text{(C8)}$$

so that

$$\frac{d\Omega}{d\eta} = -\frac{1}{\beta Z}\frac{dZ}{d\eta} = \frac{1}{\beta Z}\mathrm{Tr}\left[\frac{\partial \mathcal{G}}{\partial \eta}(\beta)\right]. \qquad \text{(C9)}$$

The variation of the Green's function $\partial \mathcal{G}/\partial \eta$ can be determined using the integral form of the Dyson equation (C7), in short form $\mathcal{L}\mathcal{G} = \mathcal{G}_0$ with $\mathcal{L} \equiv I - \eta \mathcal{G}_0 * - \mathcal{G}_0 * \Sigma *$. Taking the $\eta$-derivative gives $\frac{d\mathcal{L}}{d\eta}\mathcal{G} + \mathcal{L}\frac{\partial \mathcal{G}}{\partial \eta} = 0$ where

$$\frac{d\mathcal{L}}{d\eta} = -\mathcal{G}_0 * \left(I + \frac{\partial \Sigma}{\partial \eta}*\right) = -\mathcal{L}\mathcal{G}*\left(I + \frac{\partial \Sigma}{\partial \eta}*\right), \quad \text{(C10)}$$

so $\mathcal{L}\frac{\partial \mathcal{G}}{\partial \eta} = -\frac{d\mathcal{L}}{d\eta}\mathcal{G} = \mathcal{L}\mathcal{G}*\left(I + \frac{\partial \Sigma}{\partial \eta}*\right)\mathcal{G}$. Assuming $\mathcal{L}$ is invertible, we obtain

$$\frac{\partial \mathcal{G}}{\partial \eta} = \mathcal{G}*\left(I + \frac{\partial \Sigma}{\partial \eta}*\right)\mathcal{G} = \mathcal{G}*\mathcal{G}, \qquad \text{(C11)}$$

where in the last step have assumed $\Sigma$ is given and fixed (independent of $\eta$) when solving (C7). Inserting (C11) into (C9) then gives

$$\frac{d\Omega}{d\eta} = \frac{1}{\beta Z}\mathrm{Tr}\left[(\mathcal{G}*\mathcal{G})(\beta)\right] = -\frac{1}{\beta \mathrm{Tr}\left[\mathcal{G}(\beta)\right]}\mathrm{Tr}\left[(\mathcal{G}*\mathcal{G})(\beta)\right]. \qquad \text{(C12)}$$

We can now solve (C7) subject to the constraint (C8) by finding a root of $\Omega(\eta)$ using Newton's method. Given a guess $\eta$, we solve (C7) to obtain $\mathcal{G}$, compute $\Omega(\eta)$ and $d\Omega/d\eta$ using (C8) and (C12), respectively, and update $\eta$ using a Newton step.

## Appendix D: Solution of the pseudo-particle Dyson equation at fixed density and self-energy

The stabilization method of the previous section can be generalized to the pseudo-particle Dyson equation at a fixed density expectation value $\langle \hat{N}\rangle = N$. We explicitly write the chemical potential $\mu$ (previously absorbed into the local Hamiltonian $\hat{H}_{\mathrm{loc}}$),

$$\hat{H}_{\mathrm{loc}} \rightarrow \hat{H}_{\mathrm{loc}} + \mu \hat{N}, \qquad \text{(D1)}$$

where $\hat{N}$ is the density operator, so that the Dyson equation (C7) becomes

$$(I - \eta\, \mathcal{G}_0 * - \mu\, \mathcal{G}_0 \hat{N} * - \mathcal{G}_0 * \Sigma *)\mathcal{G} = \mathcal{G}_0. \qquad \text{(D2)}$$

The density expectation value is given by

$$\langle \hat{N}\rangle = -\frac{1}{Z}\mathrm{Tr}\left[\hat{N}\mathcal{G}(\beta)\right]. \qquad \text{(D3)}$$

We can solve the Dyson equation (D2) at a total density $\langle \hat{N}\rangle = N$ and free energy $\Omega = 0$ by varying the Lagrange multipliers $\mu$ and $\eta$. The constraints can be formulated as a two-dimensional root-finding problem

$$\mathbf{F}(\mathbf{x}) \equiv \begin{bmatrix} \Omega \\ \Delta n \end{bmatrix} = \mathbf{0}, \quad \mathbf{x} = \begin{bmatrix} \eta \\ \mu \end{bmatrix}, \qquad \text{(D4)}$$

where

$$\Delta n(\eta, \mu) \equiv N - \langle \hat{N}\rangle \equiv N + \frac{1}{Z}\mathrm{Tr}\left[\hat{N}\mathcal{G}(\beta)\right], \qquad \text{(D5)}$$

$$\Omega(\eta, \mu) \equiv \frac{1}{\beta}\log \mathrm{Tr}\left[\mathcal{G}(\beta)\right]. \qquad \text{(D6)}$$

The Jacobian is

$$J_{\mathbf{F}} = \frac{d\mathbf{F}}{d\mathbf{x}} = \begin{bmatrix} \frac{\partial \Omega}{\partial \eta} & \frac{\partial \Omega}{\partial \mu} \\ \frac{\partial \Delta n}{\partial \eta} & \frac{\partial \Delta n}{\partial \mu} \end{bmatrix}, \qquad \text{(D7)}$$

with

$$\frac{\partial \Omega}{\partial \eta} = \frac{1}{\beta Z}\mathrm{Tr}\left[(\mathcal{G}*\mathcal{G})(\beta)\right], \qquad \text{(D8)}$$

$$\frac{\partial \Omega}{\partial \mu} = \frac{1}{\beta Z}\mathrm{Tr}\left[(\mathcal{G}*\hat{N}\mathcal{G})(\beta)\right], \qquad \text{(D9)}$$

$$\frac{\partial \Delta n}{\partial \eta} = \frac{1}{Z^2}\mathrm{Tr}\left[(\mathcal{G}*\mathcal{G})(\beta)\right]\mathrm{Tr}\left[\hat{N}\mathcal{G}(\beta)\right] + \frac{1}{Z}\mathrm{Tr}\left[\hat{N}(\mathcal{G}*\mathcal{G})(\beta)\right], \quad \text{(D10)}$$

$$\frac{\partial \Delta n}{\partial \mu} = \frac{1}{Z^2}\mathrm{Tr}\left[(\mathcal{G}*\hat{N}\mathcal{G})(\beta)\right]\mathrm{Tr}\left[\hat{N}\mathcal{G}(\beta)\right] + \frac{1}{Z}\mathrm{Tr}\left[\hat{N}(\mathcal{G}*\hat{N}\mathcal{G})(\beta)\right]. \quad \text{(D11)}$$

In the last matrix element we have used the relation

$$\frac{\partial \mathcal{G}}{\partial \mu} = \mathcal{G}*\left(\hat{N} + \frac{\partial \Sigma}{\partial \mu}\right)*\mathcal{G} = (\mathcal{G}*\hat{N}\mathcal{G})(\tau), \qquad \text{(D12)}$$

and disregarded the dependence of $\Sigma$ on $\mu$ as in App. C. Similar to the previous section, we can now solve (D4) using Newton's method with the Jacobian given by (D7). At each Newton iterate we solve the Dyson equation (D2) to determine $\mathcal{G}$ for given $\eta$ and $\mu$.

## Appendix E: Self-consistent solution of the pseudo-particle Dyson equation

In Apps. C and D we have considered the self-energy $\Sigma$ to be fixed. In practice, $\Sigma = \Sigma[\mathcal{G}]$ depends on the pseudo-particle Green's function $\mathcal{G}$, so that the Dyson equation (C7) or (D2) must be solved by self-consistent iteration. As long as the final iterated solution converges, it is not necessary to solve the Dyson equation

and the root-finding problem exactly at each iteration on $\Sigma$. Therefore, instead of solving the constraint root-finding problem to convergence for each fixed $\Sigma$, we perform a single Newton update of $\eta$ and $\mu$. For example, for (D2), we obtain the following iteration:

1. Compute $\Sigma = \Sigma[\mathcal{G}]$ by evaluating self-energy diagrams.

2. Compute $\mathcal{G} = \mathcal{G}[\Sigma, \eta, \mu]$ by solving the Dyson equation (D2).

3. Update $(\eta, \mu) = \mathbf{x}$ by the Newton step

$$\mathbf{x} \leftarrow \mathbf{x} - [J_{\mathbf{F}}(\mathbf{x})]^{-1}\mathbf{F}(\mathbf{x}).$$

We perform this iteration beginning with the initial guess $\mathcal{G} = \mathcal{G}_0$, $\eta = \eta_0$, $\mu = 0$. We use a similar procedure to solve (C7), but only update $\eta$.

Little is known about the convergence properties of this self-consistent iteration, and we find in practice that the performance is problem-dependent. We therefore state several empirical observations for the $Ca_2RuO_4$ model of

Sec. V C, including limitations of our approach as well as our experience with alternative approaches. A more focused and in-depth study of the self-consistent iteration is an important topic for future research.

In general, the approach of tuning $\mu$ to fix the density becomes numerically unstable for gapped systems when the temperature is much smaller than the gap. In this case it is better to fix the chemical potential $\mu$ (in the gap) and only update $\eta$ within the pseudo-particle self-consistency. At very low temperatures, using Newton's method to determine $\eta$ becomes increasingly unstable, but this approach nevertheless provides stability to significantly lower temperatures than directly applying the approximate update rule $\eta = \frac{1}{\beta}\log\text{Tr}\,[G(\beta)]$. We also find that the numerical instabilities in the $\eta$ update step are most severe far away from the self-consistent solution, e.g., using the initial guess $\mathcal{G} = \mathcal{G}_0$ and $\eta = \eta_0$. Using a more robust update scheme like the bisection method for the first few iterations can therefore be advantageous.

We lastly note that close to the second-order phase transition, the convergence of the self-consistent iteration slows down critically. To remedy this, we have used a (Jacobian-free) quasi-Newton method for the self-energy self-consistency.

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
