# Peer review of "Automated evaluation of imaginary time strong coupling diagrams by sum-of-exponentials hybridization fitting"

_SciPost Physics_

## Round 2 · Referee Report · Nayuta Takemori (Referee 1) · 2025-7-10

Strengths

  1. Clear demonstration that the AAA-based fitting combined with bilevel optimization improves computational performance relative to DLR.
  2. Includes extensive benchmarks and comparisons, including tests on worst-case random spectra.
  3. Promising applicability to multi-orbital impurity problems within DMFT frameworks.

Weaknesses

  1. Limited discussion of the physically reliable temperature range, especially in the low-temperature regime where strong-coupling expansions are known to develop artifacts.
  2. Lack of quantitative comparison of other DMFT results to more established impurity solvers.
  3. No demonstration of scaling or error behavior in larger Hilbert spaces (n >> 3).
  4. The manuscript focuses heavily on numerical performance, with less discussion of the physical implications of the results.

Report

This is a technically executed manuscript that presents a numerically efficient and automated approach for evaluating bold pseudo-particle expansions in the Anderson impurity model. The combination of AAA-based rational approximation with bilevel optimization is elegant and convincingly demonstrated to reduce the scaling prefactor relative to previous DLR-based methods. The work shows promising applicability to multi-orbital impurity problems and includes benchmarks highlighting improved performance in terms of both runtime and accuracy. I especially appreciate that the authors have carefully benchmarked the method also against random spectra, which effectively represent worst-case scenarios in terms of hybridization complexity.

However, the manuscript could be substantially strengthened by providing a clearer discussion of the practical temperature ranges where the method yields not only numerical stability but also physically reliable results, elaborating on how the observed numerical behaviors (especially low-frequency errors and m-dependence) propagate into physically relevant observables, and clarifying the performance and error scaling in intermediate β regimes that are typical in realistic DMFT calculations. Additionally, some aspects of the presentation, including figure clarity and captions, would benefit from revision.

Therefore, I recommend publication after revision addressing the points below.

Requested changes

Major:
1. While the authors emphasize the general applicability of their method to arbitrary multi-orbital systems, the benchmarks are mainly restricted to models with two or three orbitals. The scaling of computational cost and approximation error in larger Hilbert spaces (n >> 3) is not demonstrated, and the growth of the expansion length p in such cases remains a potential limitation, particularly at large β. For example, in Fig.2, the “random poles” benchmark shows a saturation of p relative to the DLR basis size which indicates that the advantage over DLR may diminish in more complex spectral structures. Additional discussion or data illustrating this scaling would be valuable.

2. While the authors convincingly show performance at β=10 (high temperature) and β=1000 (low temperature), it would be helpful to include results for intermediate inverse temperatures, e.g., β=50–200 in Fig. 1. These are typical parameter ranges in practical applications of DMFT and quantum embedding methods. A clearer illustration of how the required number of poles p scales in this regime would strengthen the assessment of the method’s practical efficiency. It seems authors calculated for β=40 , so adding this data to Fig.1 would make the presentation more consistent and complete.

3. Fig.5 and Fig.7 demonstrate that the proposed solver can be applied within a fully self-consistent DMFT loop in a multi-orbital system, I am concerned that the observed low-frequency errors (particularly near ωₙ=0) and the clear m-dependence of the self-energy could potentially lead to inaccuracies or instability in physically relevant quantities such as quasiparticle weights or magnetic ordering. It would be valuable if the authors could comment on this by comparing the physical observables obtained in their DMFT calculations (e.g., quasiparticle weights, Neel temperatures) to results from more established impurity solvers, such as CTQMC or NRG, in order to clarify the extent to which the observed m-dependence and low-frequency errors translate into deviations in physically relevant quantities.

4. As this is primarily a methodological contribution, it would be very helpful for users if the authors could provide explicit guidance on the practical temperature range where the method can be expected to yield not only numerically stable but also physically reliable results. Since it is well known that NCA (including OCA and higher-order diagrams) and related strong-coupling expansions develop artifacts in the low-temperature regime, clarifying whether higher-order diagrams sufficiently mitigate these issues, or whether a practical β limit should be recommended, would significantly strengthen the manuscript.

5. While the manuscript provides an impressive array of numerical benchmarks and performance comparisons, I feel that it would benefit significantly from a more detailed discussion of the physical implications of the observed behaviors , exemplified by the crossing of error curves in Fig.4 (right).

Minor:
1. In Fig.1, the plotting style could be improved for readability. The points and line thickness are disproportionately large relative to the figure size, making the error curves difficult to interpret. I recommend either increasing the overall figure dimensions or reducing the marker and line widths to improve clarity. In addition, the caption of Fig.1 currently does not explicitly label the three types of spectral densities used in each column (e.g., semicircular, sum of Gaussians, sum of δ-functions), which makes the figure less accessible to the reader. I recommend adding explicit labels in the caption to clarify which spectral shape corresponds to each panel.

Recommendation

Ask for major revision

  • validity: high
  • significance: high
  • originality: high
  • clarity: high
  • formatting: good
  • grammar: excellent

Author:  Zhen Huang  on 2025-08-14  [id 5728]

(in reply to Report 2 by Nayuta Takemori on 2025-07-10)

We thank the referee for a thorough reading of the manuscript and for the constructive comments. Please see our point-by-point responses below.

  1. Exploration of models with a larger number of orbitals is an important direction of our future research. As is mentioned in the conclusion, we are in the process of incorporating Hamiltonian symmetries into the scheme, which we expect to bring systems of many orbitals (e.g., full d or f electron systems) within reach at least up to modest expansion orders, enabling a thorough exploration of the questions raised. Indeed, we consider our approach to be a promising direction for modeling systems with many strongly-correlated orbitals, for which few other methods are available. As far as we know, the error of the hybridization expansion order for systems with a large number of orbitals is not well understood, but our solver will allow us to explore this question in the future. We have added comments in the conclusion emphasizing these points. We do not agree that growth of the pole expansion order p for systems with many orbitals is a potential limitation. Indeed, in Fig. 2, the saturation of p with the number of poles in a random pole model relative to the DLR basis size suggests precisely that the advantage over DLR does not diminish for more complex spectral structures.
  2. Fig. 1 already does show a result in the intermediate temperature range beta = 50-200 requested by the referee: beta = 100. In any case, the trend of exponential convergence with rate decreasing with increasing beta (except in the case of a small discrete spectrum) is clear from the results shown.
  3. We are unfortunately confused about this comment. Our manuscript does not include any Matsubara frequency domain plots or self-energy plots. We would be happy to address this point if the referee could clarify.
  4. Our results strongly suggest the existence of two distinct regimes: (a) When the bath spectrum is gapped, we observe exponential convergence, and convergence accelerates at lower temperatures---see Fig. 4a---provided that thermal excitations remain below the gap size (see also the response to Question 5). (b) In contrast, when the bath spectrum is metallic, we observe slower convergence with increasing diagrammatic order, as expected for strong-coupling expansions. As for whether higher-order diagrams sufficiently mitigate known NCA artifacts, such as the spurious non-Fermi-liquid behavior or incorrect scaling of the Kondo temperature, these questions are more effectively addressed using a real-frequency solver. Real-time methods allow for direct comparisons with Fermi-liquid predictions or benchmark techniques like NRG; see, for example, T. A. Costi et al., Phys. Rev. B 53, 1850 (1996). A detailed comparison is beyond the scope of the present study, but we are actively developing real-frequency extensions of our approach and hope to address this question in the future. We briefly comment on the expected qualitative behavior in the conclusion.
  5. Concerning the different rates of convergence of the high temperature (blue) and low temperature (orange) calculations in Fig. 4b, we have added a comment on the origin of the qualitative different behaviors, which in this case is due to the interplay of temperature and the spectral gap of the impurity model.

Minor point 1. We have reduced the line widths in Fig. 1, and labeled the spectral densities in the caption.

---

## Round 2 · Referee Report · Jan von Delft (Referee 3) · 2025-7-14

Strengths

  1. Improved separation-of-variables algorithm for computing imaginary-time Feynman diagrams for quantum impurity models: speedup of several orders of magnitude relative to related previous work by the same authors.

  2. Clear presentation of key ideas.

  3. Several instructive benchmark examples.

Weaknesses

The code used to generate the numerical data is not publicly available yet. (But the authors did promised to publish an open source software package "in the near future").

Report

The authors address an important technical challenge: the computation of imaginary-time Feynman diagrams for quantum impurity models. They have recently published a paper (Ref. [43]) on this topic, which used a discrete Lehmann representation (DLR) to achieve a separation of variables in time-domain integrals. Here, they refine their approach by using a fitting method for the hybridization function, based on AAA rational approximation and subsequent numerical optimization. This refinement yields a computational speedup relative to their previous generic DLR approach, because the fitting scheme is not generic but tailored to the hybridization function. As a result, they achieve a speedup of several orders of magnitude. The new scheme is illustrated for several instructive benchmark examples, and for a challenging material-specific application -- a DMFT treatment of Ca2RuO4. The paper is written clearly and the main new ideas are presented in a transparent manner. The benchmark results are convincing, and the speedup relative to Ref. [43] is remarkable. Time will tell whether this becomes a standard go-to method in practice. I expect that the speedup factor achieved by this method will depend quite sensitively on the details of the hybridization function - because the costs scale as some power of p, the number of terms in their p-term expansion of the hybridization function. Therefore, in my view the authors cannot be expected to make general claims in this regard (and they don't ). But I urge them to focus on publishing an open source software package, as soon as possible. That would enable the community to try this scheme and gather experience about its performance on a case-by-case basis. I recommend publishing the paper as is, after fixing a typo in Eq. (8) (see below).

Requested changes

In Eq. (8), I believe the upper limit should be the same for both summation, i.e. m = k [according to (2.1) of Ref. [45]).

Recommendation

Publish (easily meets expectations and criteria for this Journal; among top 50%)

  • validity: top
  • significance: good
  • originality: high
  • clarity: top
  • formatting: excellent
  • grammar: perfect

Author:  Zhen Huang  on 2025-08-14  [id 5729]

(in reply to Report 3 by Jan von Delft on 2025-07-14)

We thank the referee for their positive comments and suggestions. We have corrected the typo which was pointed out. We refer to our response to Report #1 for comments on a code release.

---

## Round 2 · Referee Report · Anonymous (Referee 2) · 2025-7-14

Strengths

  1. Constructing of a numerically efficient DMFT solver for practical calculations is an important goal.
  2. The developed code, as described, is fast an effective.
  3. The text is clear and convincing.

Weaknesses

  1. The code is not available so far.
  2. The figures formt if poor.

Report

The paper presents a method for calculating bold strong-coupling diagrams in impurity models. The ultimate goal is to develop a code that performs DMFT calculations for real materials at a low numerical cost. I believe this is the right direction: while numerically exact QMC methods are valuable for fundamental and methodological research, practical calculations require efficiency comparable to that of DFT methods.

In practice, the authors introduce several improvements to existing methods. The most notable one is the use of the so-called adaptive Antoulas-Anderson algorithm. Although these improvements are conceptually incremental, their implementation provides a significant boost in productivity—the authors claim by several orders of magnitude. This result certainly deserves publication.

The paper is well-written, highly detailed, and supported by convincing examples. I do not believe it requires major revisions. However, I strongly recommend publishing the code (at least a preliminary version) alongside the paper so that readers can test it for their practical needs. Also, I would recommend to change Figures style, maybe draw less thick lines. Anyway its current appearance does not look nice for me. Once this is done, I recommend accepting the paper.

Requested changes

  1. Make code public.
  2. Work on Figure style.

Recommendation

Publish (easily meets expectations and criteria for this Journal; among top 50%)

  • validity: high
  • significance: good
  • originality: good
  • clarity: top
  • formatting: reasonable
  • grammar: excellent

Author:  Zhen Huang  on 2025-08-14  [id 5727]

(in reply to Report 1 on 2025-07-14)

We thank the referee for their positive comments. We have made modifications to the line thicknesses, also in response to Report #2. We would also assure the referee that a code will be released in the near future---we are targeting an initial release in early Fall.

---

## Round 3 · Referee Report · Jan von Delft (Referee 3) · 2025-8-20

Report

I am satisfied with the new version of this paper and recommend publication at this time, without waiting for the announced code release this fall.

Recommendation

Publish (easily meets expectations and criteria for this Journal; among top 50%)

---

## Round 3 · Referee Report · Anonymous (Referee 2) · 2025-8-21

Report

I am fully satisfied with the text. However I still think that is would be much better to publish the paper along with the code. This will not cause a big delay if the code is realised this fall.

Recommendation

Publish (meets expectations and criteria for this Journal)

  • validity: -
  • significance: -
  • originality: -
  • clarity: -
  • formatting: -
  • grammar: -

Author:  Zhen Huang  on 2025-09-19  [id 5835]

(in reply to Report 2 on 2025-08-21)
Category:
remark
answer to question

Thanks for the suggestion!

We have made the code public, see github.

The code has a detailed documentation, which includes installation guide, user-friendly tutorials and reference manual.

---

## Round 3 · Referee Report · Nayuta Takemori (Referee 1) · 2025-9-3

Report

I thank the authors for their detailed replies in the revised manuscript. Most of my previous concerns have been addressed satisfactorily. I only have one remaining point (Major point 3) :

Regarding point 3:
In my previous comment, I referred to “low-frequency errors” and “self-energy”, which may have caused a misunderstanding of the authors. What I intended to point out is based on the behaviour visible in the τ-space Green’s functions shown in Fig. 5 and Fig. 7. Since these functions are obtained from Matsubara frequency data via the sum-of-exponentials representation, my concern is that the deviations near the endpoints (τ \sim 0 and τ \sim β), may originate from inaccuracies in the low-frequency Matsubara components of the hybridization fitting especially for the magnetic ordered state. I understand that there should be no significant problem when the system is gapped and exponential convergence is observed, but I appreciate if the authors can comment on the situation in more general cases (e.g., metallic case)?

Recommendation

Ask for minor revision

  • validity: -
  • significance: -
  • originality: -
  • clarity: -
  • formatting: -
  • grammar: -

Author:  Zhen Huang  on 2025-09-19  [id 5836]

(in reply to Report 3 by Nayuta Takemori on 2025-09-03)

We thank the referee for the clarification, but still have some confusion about the updated remarks.

Fig. 7 shows physical observables (magnetizations and fillings) at different temperatures, rather than the full τ-space Green's functions. While the data is computed from Green's function values at the $\tau$ endpoints, this figure corresponds to a numerical experiment for a system beyond previous approaches, and there is no exact reference for comparison. Therefore, it does not provide a direct indication of error.

In Fig. 5, the errors near the endpoints are not particularly large. In particular, for βt = 2, 16, and 1024, the maximum errors occur in the interior of [0, β]. According to Fourier theory, inaccuracies in the low-frequency Matsubara components would produce uniform errors for all $\tau$ in $[0, \beta]$, which is not observed in our data. Consequently, the deviations at the $\tau$ endpoints are not due to low-frequency fitting errors. Rather, the errors for a calculation of a given diagram order stem from truncation of higher-order diagrams, and as we have demonstrated for both gapped and metallic cases, these errors can be systematically reduced by including higher-order contributions.

---

## Round 3 · Author Response

Dear Editor,

Please find enclosed our resubmission to SciPost Physics. We have carefully addressed all concerns raised by the reviewers and made the corresponding modification to the manuscript.

We sincerely thank you for your time and consideration.

Best regards,

Zhen Huang, Denis Golež, Hugo U. R. Strand, Jason Kaye

---

## Round 3 · List of Changes

1. Page 1 [modified]: We refer to higher-order expansions of this type as bold pseudo-particle strong coupling, or hybridization, expansions.
  2. Page 2 [added]: Another proposed approach to mitigating the sign problem, for equilibrium real time calculations, uses the bold strong coupling expansion in conjunction with diagrammatic Monte Carlo \cite{haule23}.
  3. Page 2 [modified]: Several works have used explicit expressions of the propagator along with symbolic computation, as in algorithmic Matsubara integration \cite{taheridehkordi19, elazab22, burke25, assi24}, or analytical evaluation of imaginary time integrals combined with diagrammatic Monte Carlo \cite{vucicevic20,PhysRevResearch.3.023082,kovacevic25,tupitsyn21}.
  4. Page 3 [modified]: We define the kernel \begin{equation} \label{eq:kdef} K(\tau,\omega) = -\frac{e^{-\omega \tau}}{1 + e^{-\beta \omega}}\end{equation} for $\tau \in (0,\beta)$, and by the anti-periodicity property $K(\tau,\omega) = -K(\beta + \tau, \omega) = -K(-\tau, -\omega)$ for $\tau \in (-\beta, 0)$. We will represent the hybridization function as \begin{equation} \label{eq:deltasoe} \Delta_{\nu \lambda} (\tau) \approx \sum_{l=1}^p \Delta_{\nu \lambda l} K(\tau, \omega_l). \end{equation}
  5. Page 3 [modified]: Sec.~\ref{sec:hybfit} presents an algorithm to obtain a compact SOE approximation \eqref{eq:deltasoe}, with illustrative examples for several hybridization functions. Sec.~\ref{sec:algorithm} describes the diagram evaluation algorithm in detail. Finally, Sec.~\ref{sec:results} presents several benchmark examples, including timing results.
  6. Page 4 [modified]: \begin{equation} \label{eq:barycentric} r^{(k)}(z) = \frac{n(z)}{d(z)}=\sum_{j=1}^k \frac{w_j f_j}{z-z_j} /\sum_{j=1}\DIFdelbegin \DIFdel{^m }\DIFdelend \DIFaddbegin \DIFadd{^k }\DIFaddend \frac{w_j}{z-z_j}.\end{equation}
  7. Page 6 added: sum of $\delta$-functions (left), semicircle (middle), and sum of Gaussians (right)
  8. Page 8 [modified]: multiplying by $-1$, and swapping the $\omega_l \leq 0$ and $\omega_l > 0$ summation indices to avoid overflow.
  9. Page 12[added]: The crossing of the blue and orange lines in Fig.~\ref{fig:dimer_time}(b) is a consequence of the different expansion order convergence rates of the high temperature calculation (at $\beta t = 2$) and the lower temperature calculations (e.g., $\beta t = 16$). We attribute the difference to the presence of thermal excitations across the model's spectral gap (of size $\sim t$) when $\beta = 2t$. At the lower temperatures, these excitations are exponentially suppressed.
  10. Page 15 [added]: These phases are particularly well-suited to our method, as the gapped spectrum of the hybridization function leads to exponential convergence with diagrammatic order, which is even enhanced at lower temperatures. In contrast, the correlated metallic phase, characterized by a continuous hybridization spectrum, poses greater challenges, requiring significantly higher expansion orders to achieve convergence.
  11. Page 15 [added]: This approach would enable a more thorough exploration of how the expansion truncation error behaves in systems with many orbitals, e.g. full d or f orbital shells. Extension of our method to real time diagrammatics is also possible \cite{paprotzki25,kim2025},
  12. The lines of figure 1-6 have been adjusted to be thinner.

---

## Round 4 · Author Response

Dear Editor,

Please find enclosed our resubmission to SciPost Physics. We have attached the github repository for the code associated with this paper, in which we also have detailed documentation and user-friendly tutorials.

We sincerely thank you for your time and consideration.

Best regards,

Zhen Huang, Denis Golež, Hugo U. R. Strand, Jason Kaye

---

## Round 4 · List of Changes

• Add the github repository for the code.

---

## Editorial Decision

accepted_in_target_journal